# Deep learning of cell spatial organizations identifies clinically relevant insights in tissue images

Shidan Wang [1] ✉, Ruichen Rong[1], Qin Zhou [1], Donghan M. Yang[1], Xinyi Zhang [1], Xiaowei Zhan[1], Justin Bishop[2], Zhikai Chi [2], Clare J. Wilhelm[3], Siyuan Zhang [2], Curtis R. Pickering[4], Mark G. Kris[3], John Minna [5,6,7], Yang Xie[1,8,9] & Guanghua Xiao [1,8,9] ✉

Recent advancements in tissue imaging techniques have facilitated the visualization and identification of various cell types within physiological and pathological contexts. Despite the emergence of cell-cell interaction studies, there is a lack of methods for evaluating individual spatial interactions. In this study, we introduce Ceograph, a cell spatial organization-based graph convolutional network designed to analyze cell spatial organization (for example,. the cell spatial distribution, morphology, proximity, and interactions) derived from pathology images. Ceograph identifies key cell spatial organization features by accurately predicting their influence on patient clinical outcomes. In patients with oral potentially malignant disorders, our model highlights reduced structural concordance and increased closeness in epithelial sub-strata as driving features for an elevated risk of malignant transformation. In lung cancer patients, Ceograph detects elongated tumor nuclei and diminished stroma-stroma closeness as biomarkers for insensitivity to EGFR tyrosine kinase inhibitors. With its potential to predict various clinical outcomes, Ceograph offers a deeper understanding of biological processes and supports the development of personalized therapeutic strategies.

Cells are the fundamental building blocks of tissue architecture. They are organized into cellular components, which, in conjunction with the extracellular matrix, form tissue structures[1]. Cell spatial organization refers to the arrangement, distribution, and interactions of cells within a tissue. This includes the relative positioning of different cell types, their morphological features, and the relationships among them. Cell spatial organization and tissue architecture offers critical insights into

disease states. For instance, the crosstalk between cancer cells and stromal cells is essential for invasive growth and metastasis[2,3]. Fibroblastic and necrotic cells also play significant roles in tumor proliferation and invasion[4–6]. The spatial heterogeneity of tumor-infiltrating lymphocytes is associated with the molecular profile of tumors and patient prognosis[7,8]. Examining the spatial organization of multiple cell types within tumor tissues and their surrounding

[1]Quantitative Biomedical Research Center, Peter O'Donnell Jr. School of Public Health, University of Texas Southwestern Medical Center, Dallas, TX, USA. [2]Department of Pathology, University of Texas Southwestern Medical Center, Dallas, TX, USA. [3]Department of Thoracic Oncology, Memorial Sloan Kettering Cancer Center, New York, NY, USA. [4]Department of Surgery, Yale School of Medicine, New Haven, CT, USA. [5]Hamon Center for Therapeutic Oncology Research, UT Southwestern Medical Center, Dallas, TX, USA. [6]Department of Pharmacology, University of Texas Southwestern Medical Center, Dallas, TX, USA. [7]Department of Internal Medicine, University of Texas Southwestern Medical Center, Dallas, TX, USA. [8]Simmons Comprehensive Cancer Center, UT Southwestern Medical Center, Dallas, TX, USA. [9]Department of Bioinformatics, UT Southwestern Medical Center, Dallas, TX, USA. ✉e-mail: Shidan.Wang@utsouthwestern.edu; Guanghua.Xiao@utsouthwestern.edu

microenvironments is crucial for understanding how these cells assemble and interact to produce diverse functional outcomes. Recent studies in cancer research have highlighted the importance of investigating how cell spatial organizations impact cancer biology and tumor progression[8–10]. Although recent advances in modeling cell-cell spatial interactions and their impact on biomarker expressions are noteworthy[11], many studies have predominantly concentrated on the spatial distance between two cells. This focus, although informative, may neglect the more intricate facets of cell spatial architecture, such as distinct cell types, cell structure alignment, and interactions among multiple cells. Furthermore, a prevalent approach has been to amalgamate the influences of neighboring cells indiscriminately, potentially overlooking distinct cell-cell interactions. Recent handcrafted methods[12–14] have delved into aspects of cell spatial organization. Moreover, SpaGCN[15], a graph convolutional network (GCN), integrates gene expression, spatial location, and histology to discern spatial domains and spatially variable genes in spatial molecular profiling data. In this study, we developed a GCN to decipher intricate cellular interactions at the single-cell scale. It underscores the importance of computational methods that holistically understand cell spatial organization, emphasizing cell locations, proximities, and relationships among various cell types, such as tumor cells, stroma cells, and immune cells.

Advancements in imaging technology have made whole-slide image (WSI) scanning of tissue slides a common clinical practice, producing large volumes of high-resolution digital images that contain in-depth information about tissue structures and cell interactions. Furthermore, the recent development of deep learning algorithms in digital pathology has enabled the automatic identification and classification of millions of cells from WSIs[16,17]. Studying cell spatial organizations from these routinely available WSIs may enhance our understanding of how cells are spatially organized to perform biological functions and how this organization influences tumor histology, progression, and treatment response. Convolutional Neural Networks (CNNs) have been developed for tumor detection, subtype classification, and mutation status prediction[18] based on WSIs. However, existing image-based CNNs do not specifically focus on cell spatial organization features; instead, they work with raw images that include a mix of tissue structures and external factors, such as inconsistent staining and scanning conditions. Consequently, interpreting image features and identifying cell-cell interactions that predict patient clinical outcomes become difficult. Traditional deep learning models present further hurdles for interpretation by biologists or clinicians. The mathematical outputs of these models often lack inherent meaning, creating a need to translate these numerical vectors into diagnostic or prognostic models that humans can understand. This disconnection between accurate clinical outcome prediction and the identification of influential factors underscores the need for innovative deep learning structures. By explicitly modeling cell spatial organization, we can significantly enhance the interpretability of deep learning models. This advancement would not only deepen our biological understanding but also improve the efficiency and stability of our analyses.

In this study, we developed Ceograph, a cell spatial organization-based GCN specifically designed to handle cell spatial interaction information in tissue images. Ceograph utilizes a graph model to represent cell spatial organization and incorporates features related to nuclei morphology and distribution, such as unique cell-cell interactions for different types, relative locations, and structural similarities. By adopting this approach, the Ceograph method emulates the cognitive process of the human brain, which first identifies cell types, then takes into account cell-cell spatial interactions, and finally interprets cell spatial organization, all while excluding potential noise. By assessing the relationship between individual cell morphology and interactions with diagnosis and clinical outcomes, Ceograph highlights image features that correlate with potential biological significance. This offers a starting point for further exploration into biological and clinical interpretations.

We introduce Ceograph, a GCN-based approach designed to decipher the complex interplay between cell morphologies and their spatial interactions, and its potential utility in diverse clinical contexts. With an increasing demand for methods that can characterize cell spatial organization and predict clinical outcomes using tissue images, Ceograph aims to bridge this gap. The versatility of this method is explored across three critical applications: (1) lung cancer subtype classification, (2) assessing the risk of malignant transformation in patients with oral potentially malignant disorders (OPMD), and (3) predicting treatment response in lung cancer. A unique strength of Ceograph lies in its potential for interpretability. For instance, in the context of lung cancer subtypes, we aim to see if Ceograph can capture the nuanced differences in cell organization between lung squamous cell carcinoma (SCC) and lung adenocarcinoma (ADC). Similarly, with OPMD patients, it might be possible to discern if disruptions in epithelial strata structure have implications for malignant transformations. Lastly, for lung cancer treatment response, understanding cellular features such as tumor nuclei morphology and stroma-stroma interactions could offer insights into treatment sensitivities. This work represents an interpretable GCN model to characterize cell spatial interaction for clinical outcome prediction and identification of cell spatial organization features that predict clinical outcomes using tissue images.

## Results

### A cell spatial organization-based tissue image analysis pipeline

Ceograph leverages cell spatial organization to model the impact of individual nuclei and cell-cell spatial interactions on biological phenotypes and clinical outcomes. To explicitly characterize cell spatial organization, we employed our Histology-based Digital (HD)-Staining model[16] to locate, segment, and classify each cell nucleus within a tissue image patch. Additionally, based on the segmentation results, we extracted well-defined nucleus morphology features, including size, shape, and orientation. Through this process, we transformed a tissue image into a spatial map of cells, where each cell's nucleus location, cell type, and morphology are directly accessible for the deep learning architecture. Furthermore, individual cell-cell spatial interaction features are assessed as interaction types (determined by the type of a pair of nuclei), structural concordance (quantified as parallelism between a pair of nuclei), and interaction strength (quantified as nuclei spatial closeness). All this information serves as input for Ceograph (Fig. 1a). In contrast, traditional CNN models use the entire image patch as input, which includes external noise and lacks explicit cell spatial organization features.

For a specific tissue image, the Ceograph method was applied through the following steps: (1) A set of image patches were randomly sampled from Regions of Interest (ROIs), which can be either manually annotated by pathologists or automatically determined by another computational algorithm. (2) Each nucleus within each image patch was segmented and characterized. In this study, the HD-Staining algorithm[16] was used for nuclei detection and classification. However, alternative nuclei or cell segmentation algorithms could also work, as long as the cell spatial locations were determined, making Ceograph a broadly applicable model for various spatial characterization technologies. (3) A cell spatial graph was constructed to represent the cell spatial organization in each image. The computed cell locations were used to determine the graph vertices and edges, with well-defined nuclei morphological features serving as vertex features (Supplementary Fig. 1), while cell types, concordances, and distances between pairs of nuclei used as edge features. (4) Finally, a Ceograph model was trained by incorporating cell spatial organization information through three cell spatial interaction-conditioned graph convolutional (CSIGC)

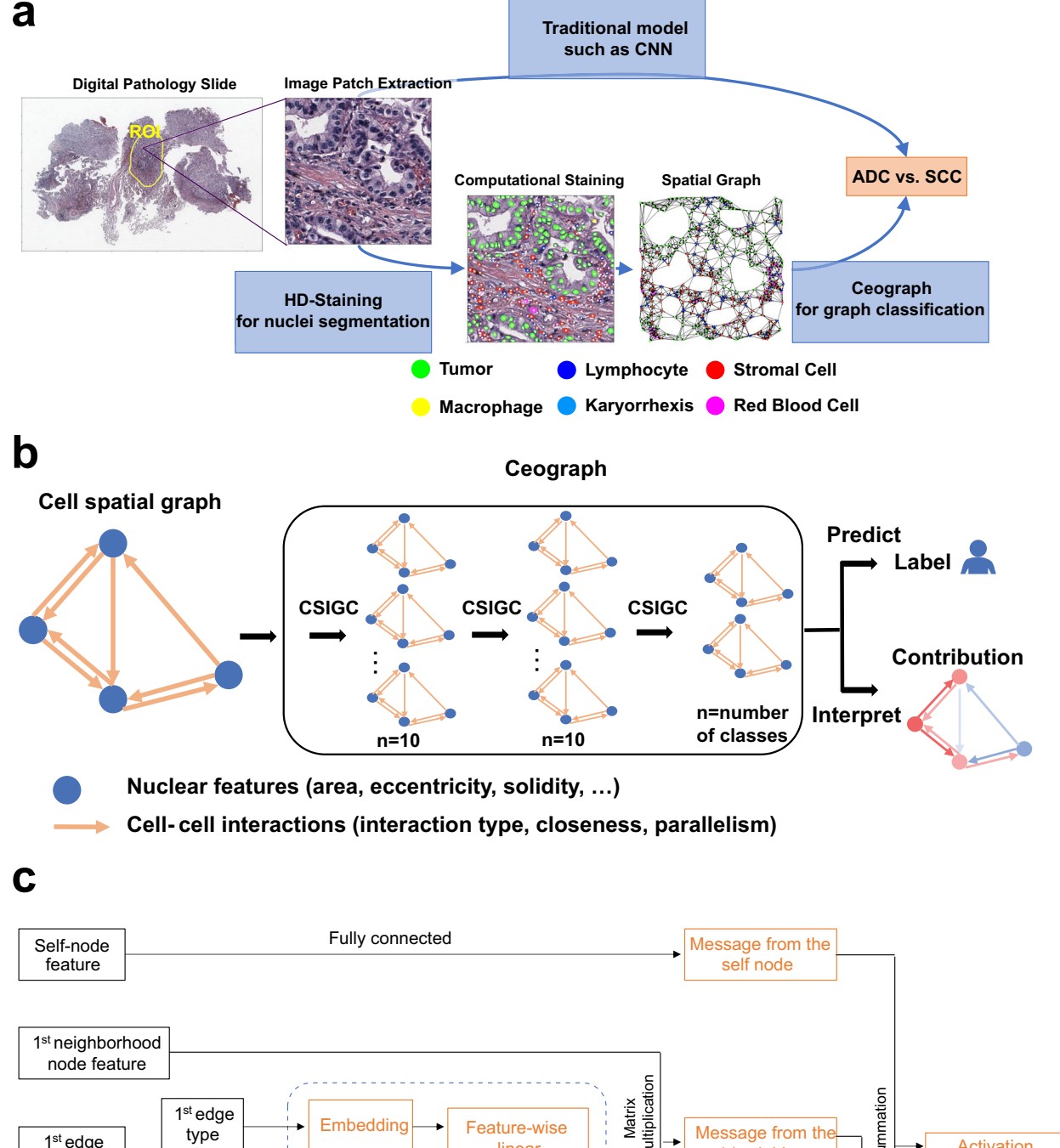

**Fig. 1 | Illustration of using Ceograph for pathology image classification.** **a** Flowchart of traditional image classification (upper arrow) and Ceograph (bottom arrow). The Ceograph method includes nuclei identification through HD-Staining, graph construction, and classification. The application to lung cancer histology subtype classification is used as an example. The image patches are 1024 × 1024 pixels under 40X magnification. **b** Ceograph structure designed in this study, enabling both classification and model interpretation. **c** Flowchart of the detailed computing process of one node feature (referred to as the self-node) through one CSIGC Layer. Orange boxes indicate the learnable layers involved in back-propagation. (ADC adenocarcinoma, SCC squamous cell carcinoma, CSIGC cell spatial interaction-conditioned graph convolution).

**Table 1 | Algorithm of cell spatial interaction-conditioned graph convolutional layer (CSIGC)**

| | Algorithm details | Description |
|---|---|---|
| 1 | Input: layer ID i, node set with N nodes, node features $X(i) \in \mathbb{R}^{N \times x(i)}$, edge features $E \in \mathbb{R}^{e \times 3}$ (three edge features are edge type $t_{n,m}$, nuclear closeness $w_{n,m}$, and nuclear parallelism $a_{n,m}$), channel number of output layer c. Output: node features $X(i+1) \in \mathbb{R}^{N \times c}$ | N: number of cells; X: nuclear morphological features; E: cell-cell interaction features. |
| 2 | for node n in N do | To describe a single cell niche |
| 3 | # Step 1: Calculate message from neighbors | Measure neighboring cells |
| 4 | for m in neighbors of node n do | With regard to each pair of the cell and one of its neighbors |
| 5 | Embedding $t_{n,m}$ to edge modulator $mo_{n,m} \in \mathbb{R}^{(x \times c)}$ through edge type embedding $EM \in \mathbb{R}^{36 \times (x \times c)}$ | Transfer categorical cell-cell interaction type into continuous features |
| 6 | Update $mo_{n,m}$ with $w_{n,m}$, $a_{n,m}$, through feature-wise linear modulation | Calculate cell-cell interaction-wise attentions |
| 7 | Reshape $mo_{n,m} \in \mathbb{R}^{(x \times c)}$ as $mo_{n,m} \in \mathbb{R}^{x \times c}$ | |
| 8 | Calculate message $me_{n,m} = X_n \times mo_{n,m}$; $me_{n,m} \in \mathbb{R}^{1 \times c}$ | Apply attention to cell features |
| 9 | # Step 2: Calculate message from the self node | Measure the cell itself |
| 10 | $me_n = X_n \times \theta^i$; $me_n \in \mathbb{R}^{1 \times c}$ | |
| 11 | # Step 3: Aggregate the messages from input layer i | Combine neighbors and self measurement together |
| 12 | Output features $X(i+1)_n = me_n + Ave(me_{n,m})$ | |
| 13 | # Step 4: Activation and training-specific dropout layer | Add non-linearity for better network flexibility |
| 14 | $X(i+1)_n = Dropout(ReLU(X(i+1)_n), training flag)$ | |
| 15 | end for | |

*ReLU* Rectified Linear Unit.

layers and another tumor cell-specific pooling layer (Fig. 1b). The model was then used to predict the category, such as differential treatment response, for each graph (constructed from an image patch). By following these steps, the Ceograph method can effectively analyze tissue images, considering cell spatial organization and interactions to deliver valuable insights for a range of clinical applications.

In the Ceograph model, one CSIGC layer was designed to incorporate cell spatial organization by combining nuclei morphological features with their spatial interactions (Table 1 and Fig. 1c). The CSIGC layer employs an attention mechanism that incorporates cell-cell spatial interaction information alongside nuclei morphological information, enabling Ceograph to focus on crucial cell-cell spatial interactions. Furthermore, the CSIGC layers integrate information from neighboring nuclei while preserving cell-cell connection structures. This approach allows for a focus on nuclei of interest (such as tumor nuclei) in the pooling layer and contributes to Ceograph's interpretability. By incorporating both morphological and spatial interaction features, the CSIGC layer enhances the model's ability to analyze complex tissue images and derive meaningful insights from cell spatial organization.

**Ceograph accurately classifies pathology subtypes in testing and independent external validation datasets**

We evaluated Ceograph's performance against traditional image-based deep learning models in a classification task for lung ADC vs. SCC pathology images. For each lung cancer WSI, we randomly sampled 1024 × 1024-pixel image patches (with 0.25 microns per pixel) from pathologist-annotated tumor regions. Using TCGA lung cancer datasets (see Dataset Section), we created training, validation, and testing sets containing 52,189 image patches from 593 slides, 7328 image patches from 83 slides, and 15,442 image patches from 172 slides, respectively (Supplementary Fig. 2), with at least 20 tumor cells per patch. The independent National Lung Screening Trial (NLST) dataset, with 36,400 graphs from 496 slides, served as an external validation dataset.

We first analyzed each image patch using HD-Staining to segment and classify six nuclei types: tumor nuclei, stroma nuclei, lymphocyte nuclei, red blood cells, macrophage nuclei, and karyorrhexis (see the Methods Section). We then derived cell spatial graphs from the HD-Staining analysis results (see the Methods Section) and trained a

Ceograph model to predict lung cancer subtypes in the TCGA training set. We systematically assessed Ceograph's performance in ADC vs. SCC classification tasks for both the TCGA testing dataset and the independent NLST dataset at image-patch and slide levels. In the TCGA testing dataset, patch-level accuracy reached 94.5% with an Area under Curve (AUC) of 0.986; slide-level accuracy achieved 100% with an AUC of 1.000 (Supplementary Fig. 3 and 4A). In the NLST external validation dataset, patch-level accuracy was 93.2% with an AUC of 0.976, while slide-level accuracy was 99.0% with an AUC of 0.999 (Fig. 2a, b and Supplementary Fig. 4A). In the lung cancer pathology subtype classification task, Ceograph outperformed the other CNN-based deep learning models that were investigated as part of this work and traditional feature-based machine learning methods (Table 2).

To further evaluate Ceograph classification performance, a CNN model with the popular ResNet101 architecture[19] and a logistic regression model were trained and evaluated on exactly the same image patches, respectively (Table 2). As expected, ResNet101 performed similarly to previous reports (0.966 slide-level AUC, Supplementary Fig. 4B). Moreover, it is noteworthy that the logistic regression model using the image patch-level averaging of image features also achieves a comparable performance to CNN-based deep learning models (0.914 slide-level AUC in the NLST dataset, Supplementary Fig. 4C), which validated the informative and interpretable nuclear features derived by HD-Staining. Importantly, the superior performance of Ceograph indicated that the cell organization information used in the cell spatial graph was critical to improving tumor subtype classification accuracy (Table 2). The direct comparison of different prediction methods once again demonstrated the superior performance of Ceograph, which benefits from explicit usage of cell spatial organization information based on histopathological knowledge.

**Understanding the prediction mechanism of ceograph**

Neural networks are frequently criticized as a "black box" due to their complex structures and massive number of parameters (up to 44.5 million[20]). Therefore, it is both challenging and intriguing to understand how a deep neural network works. The Ceograph method uses graph structure by modeling each cell nucleus as a vertex and a pair of neighboring cell nuclei as an edge. Ceograph represents cell nuclei morphological features as vertex features, while incorporating cell

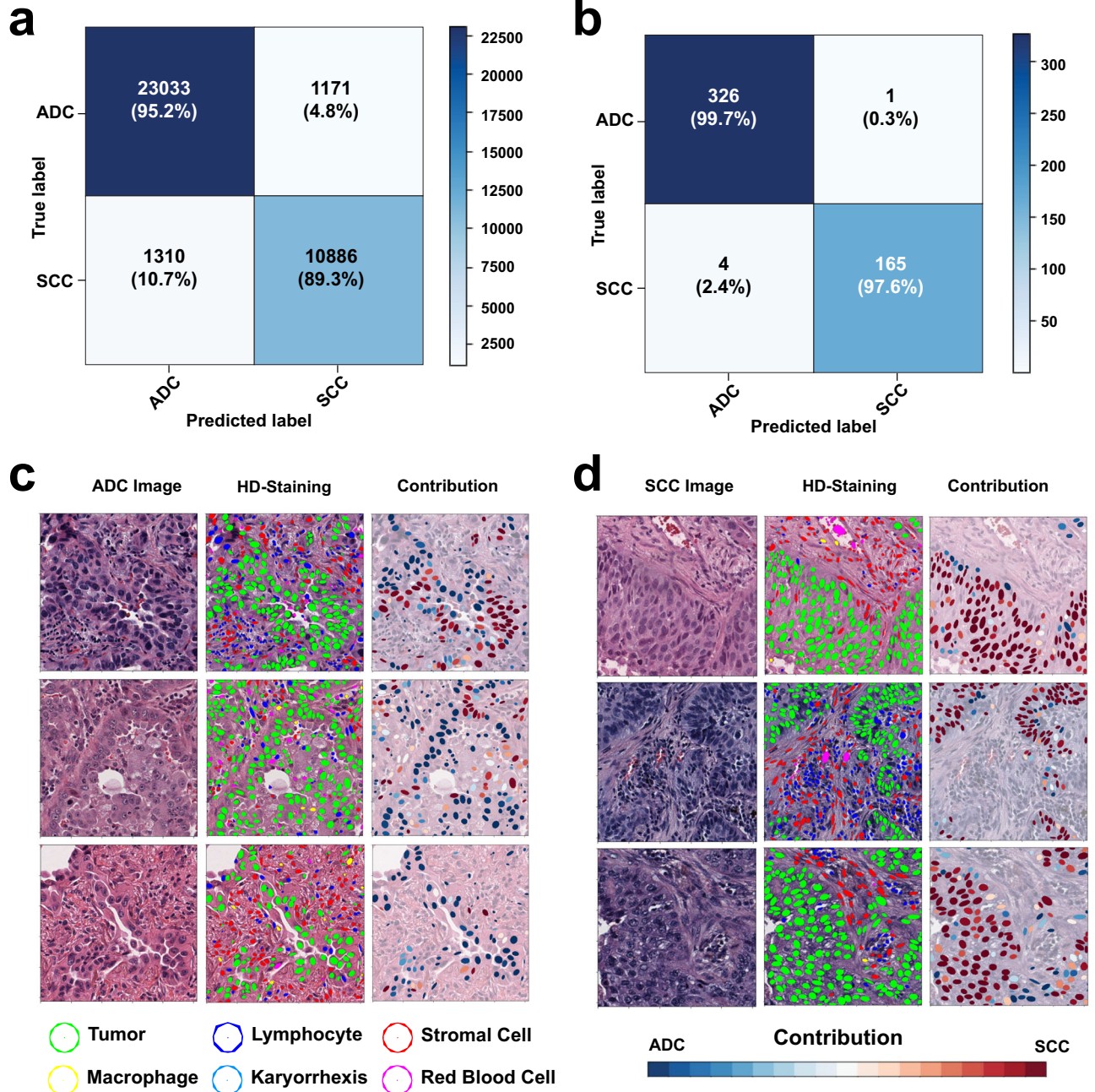

**Fig. 2 | Performance of Ceograph in lung histology subtype classification.**
Classification performance in the NLST independent testing dataset: image patch-level confusion matrix (**a**), slide-level confusion matrix (**b**). **c**, **d** Visualizing the contribution of each tumor nucleus to the final histological classification. Three examples from ADC (**c**) and SCC (**d**) are selected, respectively. Original image (1st column), HD-Staining (2nd column), and cellular-level contribution to subtype classification (3rd column) are plotted. For better illustration of nuclei features, the mask of each cell is reconstructed from nuclei morphological features extracted by HD-Staining. Blue: contributing to ADC subtype; red: contributing to SCC subtype. (ADC adenocarcinoma, SCC squamous cell carcinoma, NLST National Lung Screening Trial).

types, postures, and distances between nuclei as edge features. The contributions of these vertices and edges are then summarized through the use of CSIGC layers within the neural network. As a result, Ceograph effectively harnesses the intricate and nuanced patterns in cell spatial organization to characterize tissue images and to improve prediction accuracy. By modeling image-derived features, we visualized cellular level contributions to Ceograph's classification of ADC/SCC to understand how different cell spatial organization patterns affect the histological determination by Ceograph. The contribution of each cell to the ADC/SCC classification was calculated (detailed in the Methods) and depicted in Fig. 2c, d. Notably, regions with sheets of

polygonal cells contributed to SCC classification, reflecting the histo-pathological feature of the SCC subtype[21]. As a result, the cell-level contribution heatmap effectively highlights typical cell organization patterns for ADC and SCC.

To better understand how Ceograph makes predictions, we analyzed the impact of different cell distribution features. We visualized the contribution of each feature to the final classification of ADC vs. SCC tumors in Fig. 3a, b and found that higher values of eccentricity and solidity of tumor cells in both ADC and SCC graphs (Fig. 3a, b) increased the likelihood of a region being classified as SCC. This aligns with our prior knowledge that SCC tumor cells have higher eccentricity

**Table 2 | Performance comparison of different lung ADC vs. SCC classification models**

| Model | Training Set | Testing set | Slide-level AUC[a] |
|---|---|---|---|
| Bagging of multiple traditional machine learning models[28] | TCGA training set | TCGA testing set | 0.75 |
| Inception V1 at 40X[30] | TMA and TCGA training set | TMA and TCGA testing set | 0.83 |
| ResNet[31] | TCGA training set | TCGA testing set | 0.857 |
| Inception V3 at 20X[32] | TCGA training set | TCGA testing set | 0.87 |
| Inception V3 at 20X[29] | TCGA training set | NYU-LMC | 0.905 |
| Logistic regression using HD-Staining derived image features[b] | TCGA training set | NLST | 0.914 |
| Inception V3 at 5X[29] | TCGA training set | NYU-LMC | 0.933 |
| PathCNN at 20X[33] | TCGA training set | TCGA testing set | 0.957 |
| ResNet at 40X[b] | TCGA training set | NLST | 0.966 |
| Ceograph[b] | TCGA training set | NLST | 0.999 |

*ADC* adenocarcinoma, *AUC* area under curve, *SCC* squamous carcinoma, *NLST* national lung screening trial, *NYU-LMC* New York University Langone Medical Center dataset, *TCGA* the cancer genome atlas, *TMA* tissue microarrays.

[a]Reported AUCs in independent testing dataset are recorded (if no independent testing result is reported, testing result in the same dataset is used in the table instead). If multiple AUCs are reported, average value among all datasets is used. For the purpose of comparable comparison, in reporting AUC of the Inception V3 model[29], only frozen sections and formalin-fixed paraffin-embedded (FFPE). sections are included.

[b]Same data settings are used as direct comparison for Ceograph.

and solidity than ADC tumor cells (Supplementary Fig. 5). Additionally, we observed that the contribution from other cell types was considerably lower (both *p*-values < 0.001, Fig. 3c). Furthermore, we discovered that cell-cell spatial interaction features, such as nuclear parallelism (Fig. 3d) and nuclear closeness (inverse of distance between nuclei centroids as described in the Method Section), were critical in predicting the pathological subtype. Interestingly, we found that spatial interactions among tumor cells were more significant than interactions among other cell types (all *p*-values < 0.001, Fig. 3c). These results align with pathological observations that SCC tumor cells have a more structured architecture with elongated nuclei shapes compared to ADC tumor cells, confirming the interpretability of Ceograph.

### Ceograph predicts risk of malignant transformation of OPMD

To explore Ceograph's generalizability across different tissues and clinical questions, we applied the method to a risk stratification task. Specifically, OPMD are a group of oral cavity mucosal diseases with a risk of progressing to oral SCC. Traditionally, risk assessment involves a combination of clinical and histologic evaluation. Leukoplakia, a common type of OPMD, is assigned a histologic grading score, which is supposed to correlate with its risk of progression. However, significant intra- and inter-observer variability in dysplasia grading leads to inconsistencies and uncertainties in risk assessment and treatment planning. In this study, we employed the Ceograph method to predict the risk of malignant transformation in skin tissue for OPMD patients (Fig. 4a).

We utilized the morphological characteristics and spatial organization of the four major nuclei types within skin tissue: stratum corneum, stratum basale, other strata, and non-epithelium. Subsequently, we trained a Ceograph based OPMD malignant transformation risk prediction model using the OPMD1 dataset (refer to the Method Section for a detailed description of the OPMD1 and OPMD2 datasets) to distinguish between high-risk and low-risk groups (Supplementary Fig. 6). In the OPMD2 independent testing set, the predicted high-risk group demonstrated significantly shorter time to malignant transformation (defined as cancer-free survival time, CFS) compared to the low-risk group (Fig. 4b, *p* = 0.012; high- vs. low-risk, Hazard Ratio [HR] = 3.17, 95% Confidence Interval [CI] 1.22–3.22). Moreover, the predicted risk scores (probability of being a high risk case) from this Ceograph risk model are correlated with developing cancers within 24 months (AUC = 0.915) and 50 months (AUC = 0.797, Fig. 4c). The reduction in AUC for the 50-month prediction, compared to the 24-month prediction, may be attributed to the accumulation of additional confounding factors over the longer prediction timeline. These results suggest that Ceograph risk model can identify OPMD patients at a

higher risk of cancer by discerning the differences in cell spatial organization between OPMD tissues with a high risk of progressing to cancer and those with a low risk.

To further translate the information learned by Ceograph into pathological knowledge, we visualized the contributions of cellular-level features to Ceograph's risk assessment of malignant transformation (Fig. 4d). Feature contributions were consistent across graphs for both low-risk (Fig. 4d, upper panel) and high-risk (Fig. 4d, bottom panel) patients, and are summarized in Supplementary Fig. 7. Malignant transformation risk increases with larger nuclear areas in the stratum basale, aligning with previous pathological knowledge that increased nuclear area is a characteristic of dysplasia[22]. Comparative analysis of the OPMD1 dataset also shows that the nuclear area of stratum basale (*p* < 0.001), rather than any other stratum, was significantly higher in the high-risk group than in the low-risk group (Supplementary Fig. 8). To better comprehend the impact of cell spatial interactions and organization on malignant transformation risks, we visualized the contributions from edge features (Fig. 4d) and summarized them (Supplementary Fig. 7). In epithelial strata other than the stratum basale, reduced parallelism and increased closeness were associated with higher risk, consistent with pathological observations of disrupted cellular architecture in epithelial dysplasia. Our comparison studies confirmed decreased parallelism in epithelial strata within the high-risk group, and nuclear parallelism in other strata showed the most significant difference between high- and low-risk groups (*p* < 0.001, Supplementary Fig. 8).

In summary, the distinct contributions of cells in different epithelial layers demonstrate that Ceograph accurately and interpretably captures the morphological and structural characteristics of various epithelial strata.

### Ceograph predicts EGFR TKI Ttx response

Although epidermal growth factor receptor (EGFR) tyrosine kinase inhibitor (TKI) Targeted therapy (Ttx) is considered first-line therapy for patients with metastatic lung cancer with EGFR mutation, only 60%–80% of patients with sensitizing EGFR mutations respond to such therapy[23,24]. Developing predictive models to predict patients' response to treatment before actual administration is of great clinical importance. This might help improve treatment outcomes and potentially reduce side effects. It could also offer clinicians insights to potentially avoid less effective treatments. Given Ceograph's state-of-the-art performance in histological classification and its success in risk stratification, we further evaluated the value of Ceograph in predicting response to EGFR TKI Ttx using the tissue pathology slides collected before the treatment, which is widely regarded as a more challenging

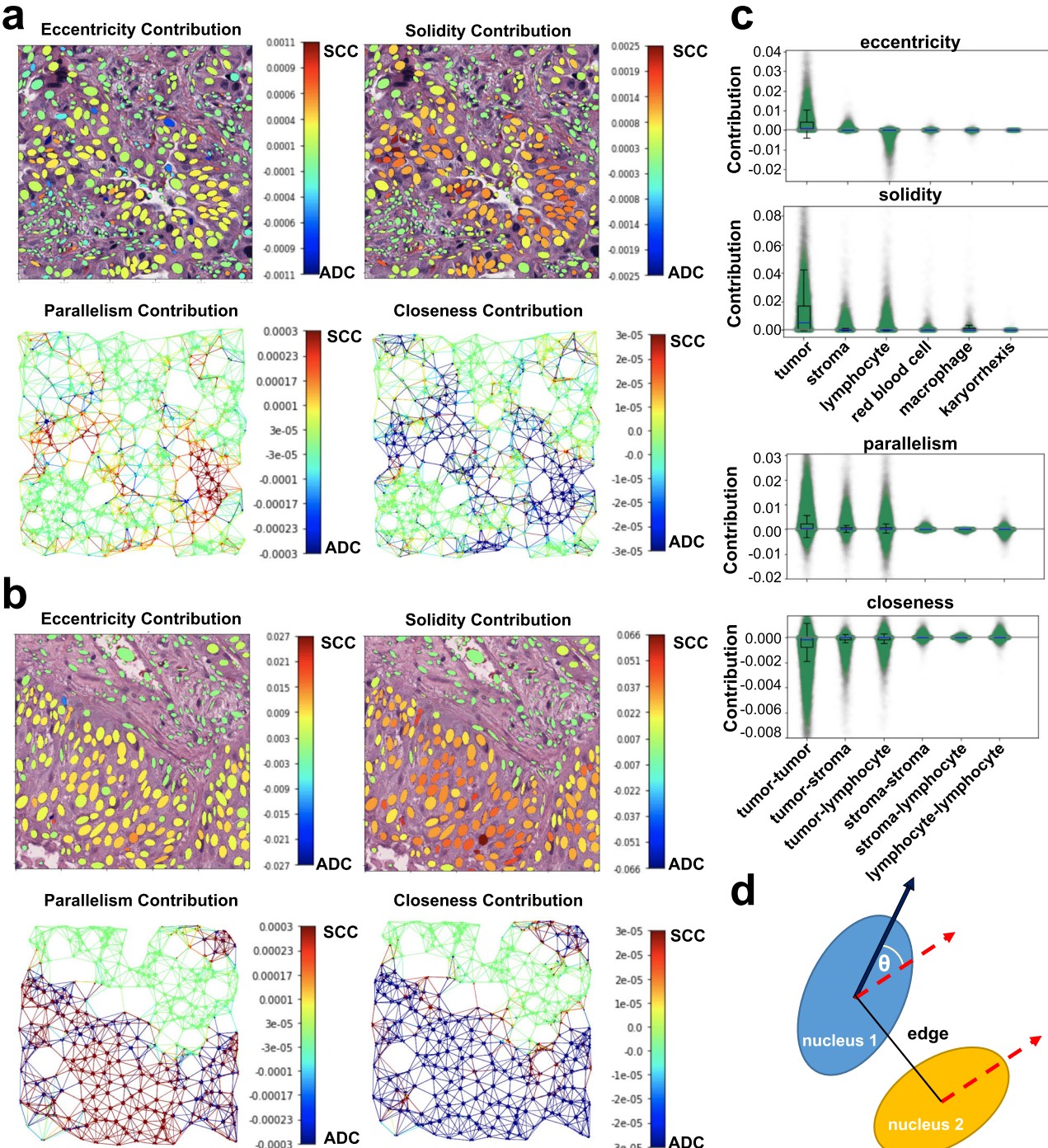

**Fig. 3 | Visualization of the working mechanism for Ceograph. a**, **b** Contributions of input features of individual cells and edges are calculated as partial derivatives of an objective function of being predicted as ADC with respect to the input features, which include both nuclei morphologies and edge attributes. The contributions are plotted for example ADC image patch (**a**) and SCC image patch (**b**), respectively. For better visualization, the contribution of edge attributes is colored in the graph edges of the upper panels. Redder color represents positive partial derivative and contribution to SCC subtype, while bluer color represents negative partial derivative and contribution to ADC subtype. **c** Boxplots to summarize feature contributions across the entire NLST dataset ($N = 229{,}157$ cells and $N = 1{,}314{,}332$ cell interactions). Positive value indicates contribution to SCC subtype, while negative value indicates contribution to ADC subtype (Statistical details included in Supplementary Data 1). **d** Illustration of the definition of "parallelism". Absolute value of Cosine θ is used to evaluate the orientation parallelism between a pair of nuclei. (ADC adenocarcinoma, SCC squamous cell carcinoma, NLST National Lung Screening).

task than risk stratification. A pipeline similar to Fig. 1a was designed and applied to the task (Fig. 5a, Supplementary Fig. 9). The predictive Ceograph was trained to predict the benefitting score for each input cell spatial graph in the Lung Cancer Mutation Consortium 1 (LCMC1) dataset where all patients with EGFR mutation were treated with EGFR TKI Ttx. Patients with overall survival (OS, defined as time from diagnosis of metastatic disease to death or last contact) >31 months (median OS time in this cohort is 31.2 months) were labeled as benefitting; otherwise, patients with OS ≤ 31 months were labeled as non-benefitting.

We applied the predictive Ceograph to patients with EGFR mutation in the independent LCMC2 testing dataset. First, the

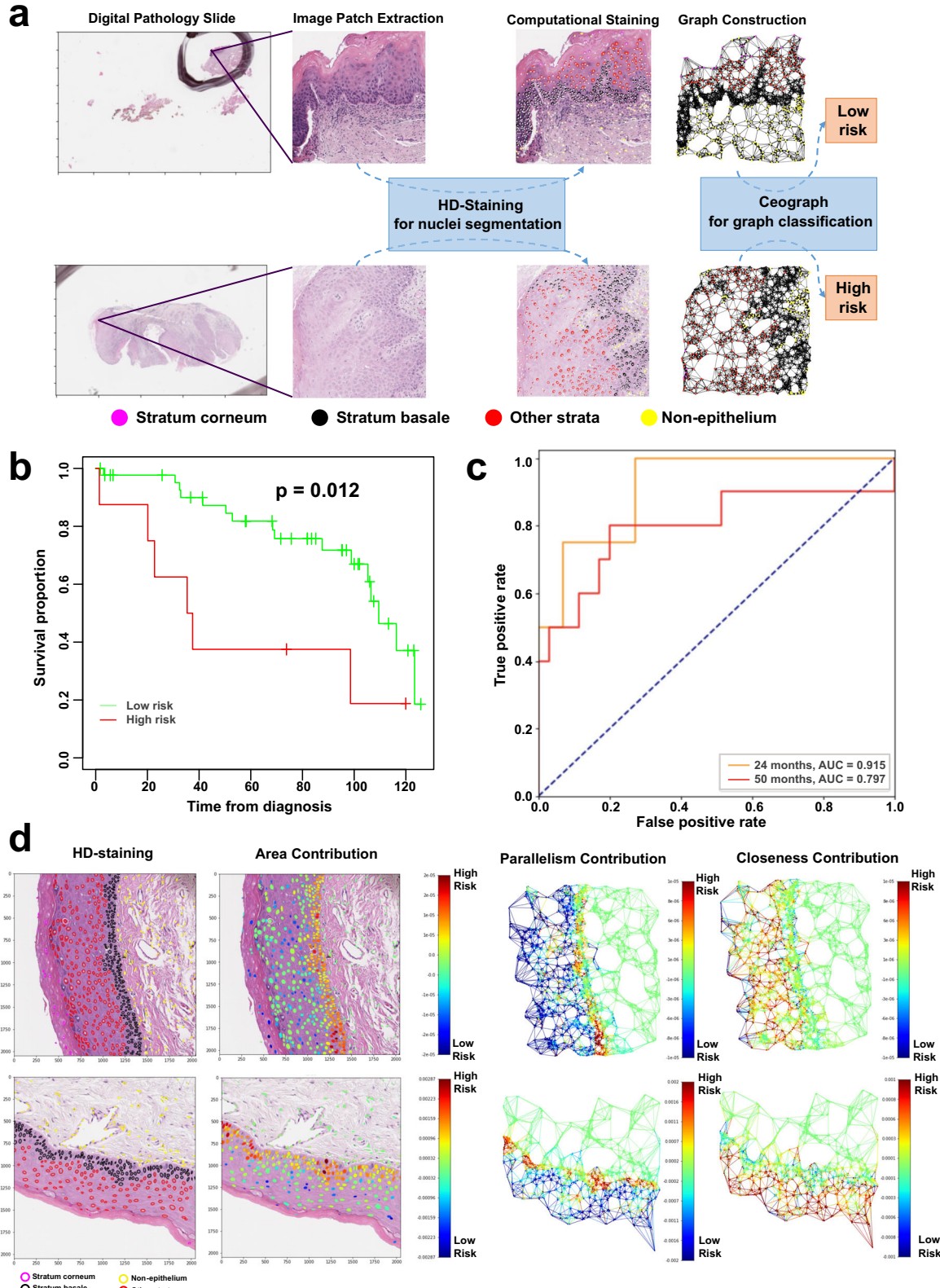

**Fig. 4 | Prognostic value of Ceograph in risk stratification of malignant transformation. a** Flowchart of risk prediction from pathology slides of Oral Potentially Malignant Disorders (OPMD) patients. The image patches are 2048 × 2048 pixels under 40X magnification. **b** Kaplan–Meier curves (Two-sided log-rank test without adjustment) of predicted high- vs. low-risk patient groups in the OPMD2 testing set. **c** ROC-curves of predicting malignant transformation events at 24 months and 50 months, respectively, in the OPMD2 testing set. **d** Model interpretation of

Ceograph via partial derivatives of an objective function of being predicted as low-risk class. Redder color represents contribution towards higher risk score with increasing the input value (e.g., area). Upper panel, an example image patch predicted as low-risk from a patient with good prognosis (cancer-free survival time CFS > 71.5 months); bottom panel, an example image patch predicted as high-risk from a patient with poor prognosis (CFS = 20.2 months).

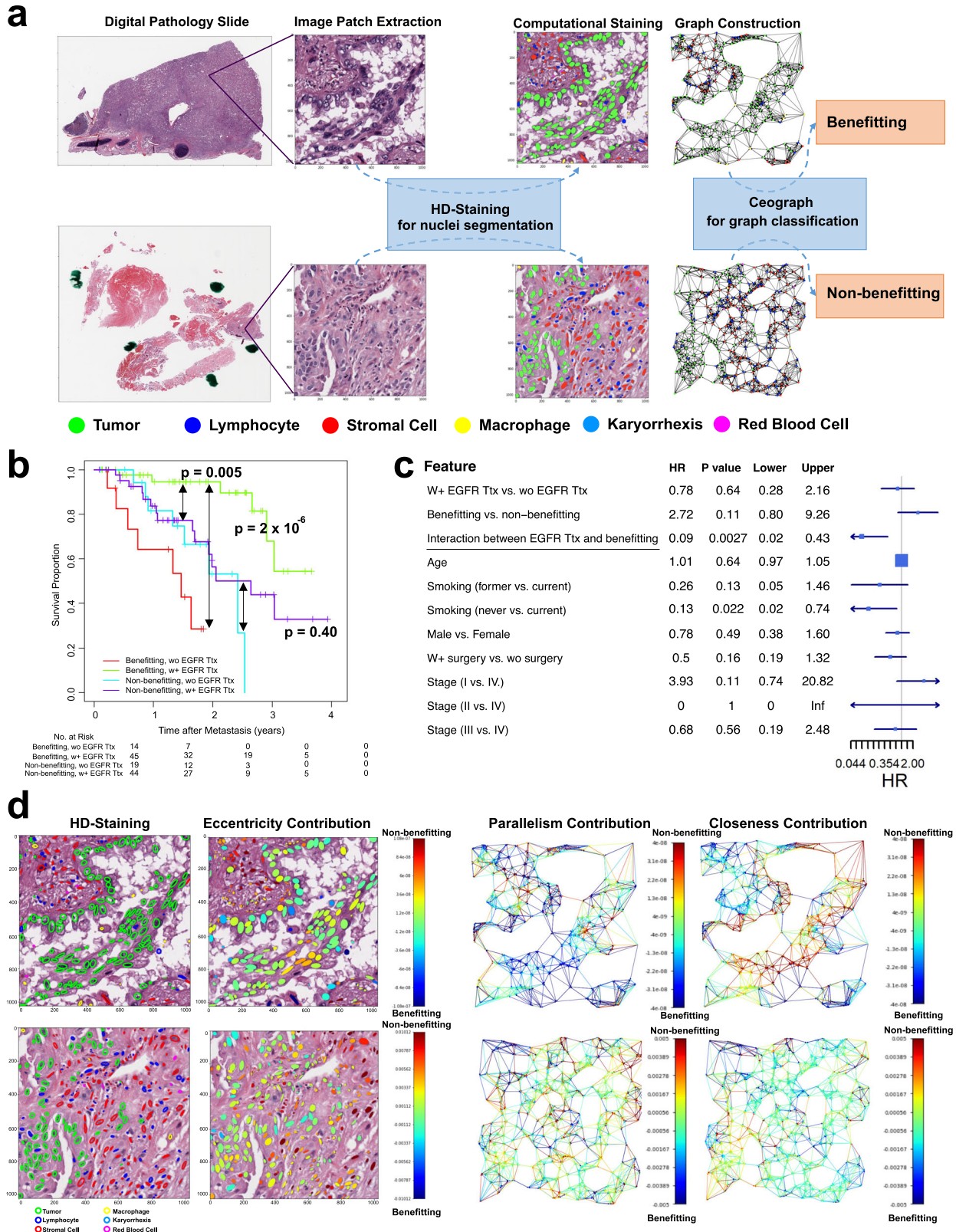

prognostic value of the predicted benefitting score was confirmed in the patients who received EGFR TKI Ttx, with the predicted benefitting group showing significantly better OS than the non-benefitting group (Fig. 5b, $p = 0.005$; benefitting vs. non-benefitting, HR = 0.30, 95% CI 0.12–0.79). More importantly, the predictive value of the Ceograph analysis was assessed by comparing the benefits of EGFR TKI Ttx in the benefitting group and non-benefitting groups. Within the predicted

benefitting group, patients who did not receive EGFR TKI Ttx showed significantly worse OS than patients who received EGFR TKI Ttx (Fig. 5b, $p < 0.001$; wo Ttx vs. w + Ttx, HR = 16.60, 95% CI 3.43–80.33). In contrast, within the predicted non-benefitting group, no significant survival difference between Ttx-treated and non-treated patient groups was observed (Fig. 5b, $p = 0.40$; wo Ttx vs. w + Ttx, HR = 1.44, 95% CI 0.59–3.49). After adjusting for potential clinical confounders,

**Fig. 5 | Predictive value of GCN in EGFR TKI Target therapy (Ttx) response prediction. a** Flowchart of EGFR TKI treatment response prediction from pathology slides of lung adenocarcinoma patients. **b** Visualize patient survival outcomes using Kaplan–Meier curves (Two-sided log-rank test without adjustment) in the Lung Cancer Mutation Consortium 2 (LCMC2) testing dataset. Patients with EGFR mutations are grouped based on their predicted benefitting score as well as whether or not they received EGFR TKI Ttx. In patients who received EGFR TKI, the significantly better survival outcome of the predicted benefitting group over the non-benefitting group ($p$ value = 0.02) demonstrates the prognostic value of the benefitting score. The improved survival outcome for patients who received EGFR TKI Ttx is detected only in the predicted benefitting group ($p$ value < 0.001) rather than the non-benefitting group ($p$ value = 0.30), demonstrating the predictive value of the benefitting score. **c** Forest plot ($N = 126$ patients) of multivariate Cox Proportional Hazard model adjusted by potential confounders, including age, smoking status, gender, surgery, and stage. **d** Model interpretation of the predictive Ceograph via partial derivatives of an objective function of being predicted as benefitting group. Redder color represents contribution of increased value (e.g., nuclear eccentricity) to decreased benefit. Upper panel, an example image patch predicted as benefitting; bottom panel, an example image patch predicted as non-benefitting. (HR hazard ratio).

including age, gender, smoking status, surgery, and stage at diagnosis, a high benefitting score calculated by Ceograph was still predictive of prolonged OS in patients with EGFR mutations who received EGFR TKI Ttx ($p = 0.0027$ for the interaction term between benefitting group and EGFR TKI Ttx, Fig. 5c). Given the heterogeneity of EGFR TKI Ttx response[23,24], our predictive model has the potential for personalized treatment selection to maximize the benefit of EGFR TKI Ttx.

To understand the relationship between cell spatial organization features and sensitivity to EGFR TKI Ttx, we interpreted the predictive Ceograph by determining how increasing individual input features would affect the predicted benefitting score. We quantified the graph feature contributions, with a positive value indicating a contribution to the non-benefitting group (Fig. 5d, Supplementary Fig. 10). Interestingly, increasing tumor cell eccentricity generally raised the likelihood of being predicted as non-benefitting from EGFR TKI Ttx. This observation aligns with comparative study results in the LCMC1 dataset, which showed significantly higher tumor cell eccentricity in the non-benefitting group compared to the benefitting group ($p < 0.001$, Supplementary Fig. 11). The observation that increasing tumor cell eccentricity generally leads to a higher likelihood of being predicted as non-benefitting from EGFR TKI Ttx suggests a connection between the elongated morphology of tumor nuclei and the epithelial-mesenchymal transition (EMT) process[25]. The EMT process has been associated with EGFR TKI-resistant tumors[26], and thus, a higher tumor cell eccentricity in the non-benefitting group may indicate EMT activity, providing a possible explanation for the observed resistance. Furthermore, the model revealed that decreased stroma-stroma closeness generally correlated with an increased likelihood of non-benefitting from EGFR TKI Ttx, consistent with observations in the comparative study ($p < 0.001$, Supplementary Fig. 11). Decreased stroma-stroma closeness may suggest the activation of cancer-associated fibroblasts (CAFs) with increased cell size, which has been linked to tumor progression and poor prognosis[27]. This implies an additional role for CAF activation in EGFR TKI resistance.

## Discussion

In this study, we developed Ceograph, a GCN-based deep learning model to characterize cell spatial organization in tumor tissues. Ceograph captures the subtle but coordinated changes in cell spatial organization, leading to state-of-the-art prediction performance in histology classification, as well as in prognostic and predictive tasks. We validated the performance of Ceograph in three applications: (1) lung cancer subtype classification, (2) malignant transformation prediction in patients with OPMD, and (3) treatment response prediction in lung cancer. Our results demonstrate that Ceograph achieves state-of-the-art performance in lung cancer subtype classification, surpassing both traditional machine learning models[28] and CNNs[29–33]. Moreover, Ceograph can predict clinical outcomes across various tissue types. We also demonstrated the interpretability of Ceograph models through the following examples: (1) In lung cancer subtype classification, Ceograph identifies features that align with existing histopathology knowledge, showing that lung SCC has a more structured architecture than lung ADC. (2) In OPMD patients, Ceograph reveals that a disruption in the structure of epithelial strata, apart from the

stratum basale, is associated with a high risk of malignant transformation. (3) In lung cancer treatment response prediction, Ceograph finds that elongated tumor nuclei and decreased stroma-stroma closeness are linked to reduced survival benefits from lung cancer targeted therapy (Ttx), providing new insights into the biological mechanisms of treatment insensitivity by demonstrating their contributions. To the best of our knowledge, this is the first study to develop comprehensive clinical outcome prediction models by characterizing cell spatial organization using graph models at the single-cell level.

Using the proposed Ceograph model, we demonstrated that cell spatial organization features, including cell spatial distribution, cell-cell interaction, and nuclei morphological features at different locations, contain sufficient information for various patient outcome prediction tasks. These tasks encompass predicting tumor histological subtypes, malignant transformation, and therapeutic response. Our findings highlight the importance of cell spatial organization in distinguishing tumor subtypes and patient outcomes.

In this study, we designed the Ceograph model to use nuclei morphological and distribution features as input based on pathological knowledge, mimicking the human brain's hierarchical image recognition and classification process. The model first "recognizes" cells using the HD-Staining model and then discerns comprehensive but subtle patterns in cell organization and morphological characteristics to classify tissue into categories associated with histology subtype or potential patient outcomes. This "cell recognition" step reduces the effect of variations in staining or lighting conditions, improving stability in the "classification" step.

The proposed Ceograph model represents a successful attempt to combine human knowledge with a neural network. As a result, the Ceograph model outperforms traditional CNN models while utilizing far fewer parameters. The proposed histology classification Ceograph, in particular, has only 9912 parameters, which is roughly 1/5000th of the popular ResNet101 model (#parameters = 44.5 million)[20] and 1/2500th of the InceptionV3 model (#parameters = 23.9 million)[34]. Decreasing the number of parameters improves portability, computational load, and interpretability while reducing the risk of overfitting. Although the Universal Approximation Theory[35] proves that neural networks can approximate any Borel measurable functions given enough hidden layers, large numbers of trainable parameters have long been contradictory with the constraints of limited training samples. Increasing signal-to-noise ratio and utilizing existing knowledge are important for resolving this contradiction, in addition to collecting and augmenting training data.

Understanding how a neural network makes a decision is important. Guided by the occlusion- and gradient-based methods, which have been utilized to estimate marginal attributes of an input feature applied to CNNs[36,37], we propose that calculating partial derivatives of output with regard to each individual input feature is a simple and effective way to understand the contribution of each feature to the decision making process of Ceograph. The consistency of knowledge generated by Ceograph and direct comparative study demonstrates the validity of this method. More importantly, the results suggest that Ceograph models offer interpretability, indicating potential

correlations in how cells assemble and interact across various contexts. This provides avenues for further exploration into their functional implications.

One limitation in this study is that the proposed Ceograph uses only nuclei features rather than cytosolic or plasma membrane features. Although cellular architecture is an important factor in the subtyping process by pathologists, cytosolic features are also helpful. For example, the cytosol morphology of signet ring ADC and mucinous ADC differs[38], and the presence of obvious plasma membrane is a marker for SCC. Although we demonstrated that the morphological features and nuclei distributions are useful for distinguishing between ADC and SCC, predicting risk of malignant transformation of OPMD patients, and predicting treatment response to EGFR TKI Ttx, combining cytosolic and membranous features may be useful in other tasks.

In the process of training our model, we employed a deliberate stratification approach: patients who developed cancer quickly were categorized as the high-risk group, while those who remained cancer-free over an extended follow-up period were designated as the low-risk group. This strategy aimed to reduce ambiguity in survival outcomes during model training. However, this method has a notable limitation: it does not fully utilize on the information from patients with censored outcomes. As highlighted by recent studies[39], coupling a neural network with a loss function specifically designed for survival analysis could potentially improve predictive performance for survival outcomes.

The edge effect is a known challenge in image analysis that focuses on ROIs, particularly with our Ceograph method that utilizes k-Nearest Neighbors for graph construction. When cells lie on the boundaries of ROIs, they're constrained to have neighbors only within that ROI. This design can introduce biases for these boundary cells. However, considering the substantial size of our typical ROIs containing a large number of cells, while the number of cells affected by the edge effect is relatively small. It is important to recognize this inherent limitation and consider it in future methodological refinements.

## Methods
### Dataset
**TCGA and NLST datasets for histology classification.** Pathology images that support the lung cancer subtype classification results of this study are available online in TCGA (https://wiki.cancerimagingarchive.net/pages/viewpage.action?pageId=6881474) and NLST (https://biometry.nci.nih.gov/cdas/nlst/). The H&E-stained pathology images (40X), together with the corresponding clinical data, were obtained from the NLST and TCGA cohorts: 469 pathology slides for 422 lung ADC patients and 379 pathology slides for 379 lung SCC patients were acquired from the TCGA Lung SCC dataset; 327 pathology slides for 193 lung ADC patients and 169 pathology slides for 93 lung SCC patients were acquired from the NLST dataset (there could be multiple pathology slides for a single patient). Dataset details are summarized in Supplementary Table 1. To refine our analysis within the tumor region, the tumor ROI for each of the pathology slides were labeled and confirmed by pathologists using the ImageScope software (Fig. 1a). All samples in TCGA have been collected and utilized following strict human subjects protection guidelines, informed consent and IRB review of protocols. All participating sites obtained local IRB approval for participation in this study. More information can be found at TCGA website (https://www.cancer.gov/ccg/research/genome-sequencing/tcga). All the TCGA imaging data are available at NCI Imaging Data Commons (https://datacommons.cancer.gov/repository/imaging-data-commons). All participating sites obtained local IRB approval for participation in this study. All samples in TCGA have been collected and utilized following strict human subjects protection guidelines and informed protocols.

**OPMD1 and OPMD2 datasets for malignant transformation prediction.** Patients with baseline OPMD were biopsied and monitored at the Department of Head and Neck Surgery at The University of Texas MD Anderson Cancer Center (MDACC). The OPMD pathology images, reviewed by expert pathologists and scanned at 40X magnification, were acquired in two distinct batches. These batches are referred to as the OPMD1 and OPMD2 datasets, respectively. The risk of malignant transformation was quantified using the Cancer-free Survival (CFS) time, defined as the duration from the diagnosis of OPMD to the diagnosis of oral cancer or the last follow-up. Patients with concurrent cancers were excluded from the OPMD datasets. For OPMD datasets, written informed consent was obtained from patients included in the study. This study was approved by the Institutional Review Board at MD Anderson Cancer Center.

In this study, for OPMD, patients who do not develop oral cancer within 60 months are categorized as low-risk, while those who develop oral cancer within 42 months are considered high-risk. Based on this criterion, and in conjunction with the set exclusion criteria, we identified 17 patients who had a follow-up duration exceeding 60 months without developing oral cancer (the shortest follow-up time among these patients was 68.5 months). On the other hand, six patients who developed oral cancer within a span of 42 months (with CFS < 42 months and an event status of true) were classified as high-risk and included in the OPMD1 training data. For validation, we used the OPMD2 dataset, we used data from 53 patients in the Erlotinib Prevention of Oral Cancer (EPOC) trial at MDACC. This group encompasses patients with a range of CFS durations and varying censoring/progression statuses. The most extended follow-up duration record ed was 126 months. Written informed consent was obtained from patients included in the study. This study was approved by the Institutional Review Board at MD Anderson Cancer Center. Written informed consent was obtained from patients included in the study.

**LCMC1 and LCMC2 datasets for EGFR TKI Ttx response prediction.** The LCMC1[40] and the LCMC2[41] datasets are multi-institutional datasets established to study different oncogenic drivers and corresponding target therapies. Patients who had stage IV or recurrent lung ADC, Southwest Oncology Group performance status 0, 1, or 2, more than 6 months of expected survival, and tissue adequate for molecular analyses were considered eligible for LCMC1 and LCMC2. Biopsy slides and clinical information for patients who carried EGFR mutation were collected for this study. 115 biopsy slides from 98 patients were collected from the LCMC1 dataset and 137 biopsy slides from 126 patients were collected from the LCMC2 dataset. Patient characteristics are summarized in Supplementary Table 2. Fourteen clinical sites participated in the LCMC (Supplementary Table 3). All participating sites obtained local IRB approval for participation in this study. Fourteen clinical sites participated in the LCMC. All participating sites obtained local IRB approval for participation in this study. Written informed consent was obtained from patients included in the study.

### Nuclei segmentation and classification using HD-Staining
Identification and classification of different cell types is a prerequisite for constructing cell graphs. An automatic deep learning model, HD-Staining, was trained to identify six different cell types in the lung cancer microenvironment: tumor cells, stromal cells, lymphocytes, red blood cells, macrophages, and karyorrhexis[42]. The model was directly applied to image patches within tumor ROIs of pathology images (Fig. 1a) and produced information about each identified nucleus, including centroid position, cell type, confidence of prediction, nuclear orientation (defined as the angle between x-axis and major axis of nucleus), and 10 well defined nuclear morphological features (area, convex area, eccentricity, extent, filled area, major axis length, minor

axis length, perimeter square divided by area, perimeter, and solidity). The model is available online as a web-server at http://lce.biohpc. swmed.edu/maskrcnn/. Since HD-Staining was trained on pathology slides at 40X magnitude, pathology images at 20X magnitude were resized to 40X using SR-GAN[43]. For 40X images, the spatial resolution is 0.25 microns per pixel (mpp).

In order to characterize the epithelial microenvironment for OPMD patients, we further generalized HD-Staining to oral epithelial tissues. The epithelial HD-Staining was trained to identify four different cell types: stratum corneum, stratum basale, other stratums, and non-epithelium. The same set of information as described before was extracted for each identified nucleus. Board-certified clinical pathologists have reviewed and approved the HD-staining results for datasets from all three examples used in this study.

## Graph construction using k-Nearest neighbors

The spatial organization of all cells within an image patch naturally fits in the graph concept. Thus, a directed graph was constructed for each image patch using k-Nearest Neighbors based on Euclidean distance with the direction pointing from a cell to its neighbors (Fig. 1a). K was set to 8 to cover the adjacent neighbors of each nucleus. Each graph consisted of two components: nodes (representing nuclei) and edges (representing spatial interactions among nuclei). To further describe cell types, morphological features, and cell-cell spatial interactions as a graph, two feature matrixes were defined for nodes and edges, respectively. The node feature matrix contained 11 features: confidence of prediction and the 10 aforementioned morphological features generated by HD-Staining. The node features were globally centered and scaled before being fed into the Ceograph model. The edge feature matrix contained three features: categorical edge type based on cell types of starting node and ending node (yielding $6*6=36$ edge types for 6 cell types in lung cancer microenvironment and $4*4=16$ edge types for 4 cell types in epithelial microenvironment), nuclear parallelism defined as the absolute value of cosine of angle between major axes of the starting node and the ending node (a greater value indicates greater parallelism, Fig. 3d), and nuclear closeness defined as the reciprocal of Euclidean edge length in pixels.

## Lung ADC vs SCC classification using Ceograph

We hypothesized that instead of directly using images, which are high-dimensional, a much simpler graph representation of cell spatial organization and nuclei orientation would be informative enough to distinguish the two main histological subtypes of lung cancer, ADC and SCC. Thus, Ceograph, a GCN, was designed to utilize graph data instead of a CNN, which is only able to analyze structured data (Fig. 1b). Ceograph was constructed with three cell spatial interaction-CSIGC layers[44] followed by a Subgroup Mean Pooling layer and a Softmax layer. The Ceograph consisted of nodes (nuclei) and edges (representing their spatial interactions). For each node, 11 nuclear morphological features were utilized as input data: nuclear area, convex area, eccentricity, extent, filled area, major axis length, minor axis length, the ratio of perimeter squared to area, perimeter, solidity, and prediction confidence. For each edge, three features were used: the categorical edge type determined by the cell types of both the starting and ending nodes resulting in a total of ($n \times n$ types for n cell types), nuclear parallelism, and nuclear closeness. The computation algorithm of a CSIGC layer is illustrated in Fig. 1c and Table 1. The interaction-conditioned graph convolution makes the convolution operation conditioned on the spatial interactions (i.e., edge attributes). Through the Subgroup Mean Pooling layer, only features for tumor nuclei nodes were averaged to focus on contributions from tumor nuclei.

The classification Ceograph model was trained, validated, and tested using the TCGA dataset, and independently tested in the NLST dataset. To construct the training/validation/testing datasets, 100

1024 × 1024 pixels image patches were extracted from the ROI of each pathology slide (Supplementary Fig. 2) and transformed into 100 graphs individually; only graphs containing at least 20 tumor cells were considered as informative enough to be classified as ADC/SCC and kept in the datasets. The TCGA ADC dataset contains 40,971 graphs (28,522 graphs from 328 slides were assigned to the training dataset, 3985 graphs from 46 slides were assigned to the validation dataset, and 8464 graphs from 95 slides were assigned to the testing dataset); the TCGA SCC dataset contains 33,998 graphs (23,667 graphs from 265 slides were assigned to the training dataset, 3353 graphs from 37 slides were assigned to the validation dataset, and 6978 graphs from 77 slides were assigned to the testing dataset). Graphs from the same slide were assigned to the same dataset to avoid data leakage. The independent NLST testing dataset contains 24,204 graphs from 327 ADC slides and 12,196 graphs from 169 SCC slides.

To train the ADC vs. SCC classification Ceograph, cross-entropy was used as loss function; Stochastic Gradient Descent (SGD) with learning rate = 0.0001 and momentum = 0.9 was used as optimizer. The model with highest classification accuracy in the validation dataset was selected and applied to the testing datasets. Majority voting among all graphs from labeled ROI was used to determine the histology subtype of a slide. Both patch-level and slide-level accuracies and receiver operating characteristic (ROC) curves were calculated to evaluate the classification performance.

## Ceograph for malignant transformation prediction

Given the limited sample availability for OPMD malignant transformation research, our modeling strategy relied on a select set of cases with clear clinical phenotypes as the training dataset (OPMD 1 data) and a larger, independent cohort (OPMD 2 data) mirroring real-world clinical scenarios for validation. The training data utilized a binary outcome—categorized as high-risk or low-risk. A Ceograph-based classification model was developed to predict a new case as a high risk or low risk case. The prediction performance was then validated in the independent cohort by examining the association between predicted risk groups and the observed time to cancer using K–M curves and log-rank tests. CFS time and event status (progress to oral cancer or not) were used to quantify risk of malignant transformation and defined as time from diagnosis of OPMD to diagnosis of oral cancer, or last follow-up.

Patients at MDACC with OPMD were biopsied and monitored. Within the OPMD1 dataset, 17 patients who did not develop cancer over a span of 60 months were tagged as low-risk, while six patients who exhibited cancer signs within 42 months were considered high-risk. The OPMD2 dataset, derived from 53 patients involved in the Erlotinib Prevention of Oral Cancer trial, was designated for validation, comprising patients with diverse CFS durations (refer to the Dataset Section). During each training epoch, a unique 2048 × 2048 pixel image patch containing a minimum of 50 epithelial nuclei was randomly chosen from each patient. Hence, image patches from the same patient varied across epochs. For the testing set, 100 distinct 2048 × 2048 pixel image patches were randomly selected from each OPMD2 dataset patient, ensuring each patch had at least 50 epithelial nuclei. Image patches from both OPMD1 and OPMD2 were each transformed into individual graphs.

To access the utility of Ceograph in predicting risk of malignant transformation of OPMD patients, we adapted the pathology image analysis pipeline proposed in Fig. 1a to oral epithelial tissues (Fig. 4a). To increase the receptive field of Ceograph, we increased the number of CSIGC layers to 4; to focus on epithelial cells, only features for epithelial stratum nuclei nodes were averaged through the Subgroup Mean Pooling layer (Supplementary Fig. 6).

To train the malignant transformation risk prediction model, cross-entropy was used as loss function; SGD with learning rate = 0.0005 and momentum = 0.9 was used as optimizer. The model with highest

classification accuracy in the OPMD1 set was selected. The probability of the graph predicted as high-risk group was used as the risk score of malignant transformation. The graph-level risk scores were averaged across all 100 patches for each individual patient to be summarized into patient-level risk scores. Patients were dichotomized into high- and low-risk groups with the patient-level risk score cutoff = 0.5.

## Ceograph for EGFR TKI Ttx response prediction

To assess the utility of Ceograph in predicting treatment response to EGFR TKI Ttx in patients with lung ADC, a deep learning-based pathology image analysis pipeline similar to Fig. 1a was used (Fig. 5a). To adapt for patient-level treatment response, the pipeline was modified in the following ways: (1) the categorical outputs were defined as non-benefitting and benefitting groups; (2) all connected graphs from one pathology slide were included in a single disconnected graph; (3) the subgroup mean pooling layer was replaced with a global mean pooling layer to take all cell types within the tumor microenvironment into consideration (Supplementary Fig. 9).

The predictive Ceograph model was trained on the LCMC1 dataset and independently tested on the LCMC2 dataset. OS, defined as time from diagnosis of metastatic disease to death or last follow-up, was used as the outcome. Within the LCMC1 dataset, patients who died within 31 months (OS ≤ 31 months and event = true) were categorized as non-benefitting, while those who survived longer than 31 months (OS > 31 months) were classified as benefitting (Supplementary Table 2). The cutoff was selected based on the median OS time, which is 31.2 months, to ensure that the non-benefitting and benefitting groups are relatively balanced.

To train the response prediction Ceograph model, cross-entropy was used as the loss function; AdaDelta[45] with scaling factor = 2 was used as the optimizer. The model with the highest classification accuracy in the LCMC1 dataset was selected. The probability of belonging to the benefitting group was used as the benefitting score. In the testing dataset, the benefitting scores were averaged into patient-level; then, the patients were dichotomized into benefitting and non-benefitting groups according to the median benefitting score.

## Ceograph interpretation

The working mechanism can be interpreted in two aspects. First, since graph level supervision works as cellular level weakly-supervision, cellular level predictions can be computed to understand local contribution to the global prediction:

$$x_{graph,k} = \frac{1}{N}\sum_{i=1}^{N} x_{nucleus,i,k} \forall k \in 1, 2, \ldots, K \quad (1)$$

$$P_{graph} = Softmax\left(x_{graph}\right) \quad (2)$$

$$P_{nucleus,i} = Softmax(x_{nucleus,i}) \quad (3)$$

where $K$ is the number of prediction categories, $N$ is the number of nuclei involved in the Subgroup Mean Pooling Layer, and $P$ represents the predicted probability vector. Mapping and visualizing $P_{nucleus}$ spatially enabled understanding of how local cell spatial organization patterns affect the whole graph determination (Fig. 2c, d).

Second, feature-wise contribution to the graph prediction can be computed in a model agnostic way[37]:

$$Contribution_f = \frac{\partial L}{\partial f} \quad (4)$$

$$L = Cross Entropy(P_{graph}, Presumptive category k) \quad (5)$$

where f represents either node (nucleus) or edge (cell-cell spatial interaction) feature of an individual node or edge, and L is an objective (loss) function. Hereby, partial derivative is used to conveniently calculate how increasing f by a small amount affects the objective function. A positive partial derivative indicates that increasing f results in a larger loss value; thus, feature f contributes negatively to the category k in this case. In practice, the partial derivative for all features could be calculated simultaneously, making this strategy a quick and efficient way to assess the contribution of each derivable feature. Mapping the node or edge level contributions back to the spatial locations enabled understanding of how local cell organizations contribute to the graph level prediction. A boxplot grouped by different nucleus or cell-cell connection types and across an entire dataset allowed for further comparison with pathological knowledge and identification of the key nucleus or cell-cell connection types and features that could be important biomarkers.

## Survival analysis

Kaplan–Meier (KM) curves, log-rank tests, and Cox Proportional Hazard (CoxPH) models were used to evaluate the survival difference between two patient groups: between high- and low-risk groups to evaluate the performance of Ceograph risk model in predicting risk of malignant transformation, and between EGFR TKI Ttx treated and non-treated patients in the benefitting group and non-benefitting group, respectively, to evaluate performance of predictive Ceograph in predicting response of EGFR TKI Ttx. ROC curves were plotted and AUC scores were calculated to evaluate the performance of the risk stratification Ceograph model in predicting 24- and 50-months cancer-free-survival probability for patients with OPMD, respectively. Multivariate CoxPH model was used to adjust for potential confounders, including age, gender, smoking status, surgery, and stage at initial diagnosis. The differences were considered significant when the two-tailed p-value was less than 0.05. Python libraries "scikit-learn" (v0.20.3), "torch" (v1.0.1.post2)[46], "torch-geometric" (v1.0.3) were used for graph creation and Ceograph implementation; the R library "survival" (v2.41-3) was used for survival analysis.

## Reporting summary

Further information on research design is available in the Nature Portfolio Reporting Summary linked to this article.

## Data availability

Pathology images that support the findings of this study were available online in NLST (https://biometry.nci.nih.gov/cdas/nlst/) and The Cancer Genome Atlas Lung Adenocarcinoma (TCGA-LUAD, https://wiki.cancerimagingarchive.net/pages/viewpage.action?pageId=6881474). The LCMC1 and LCMC2 datasets are sourced from the Lung Cancer Mutation Consortium (LCMC). The LCMC1 and LCMC2 datasets are controlled access. Controlled access to the LCMC datasets is implemented to protect the privacy and confidentiality of research participants and to comply with ethical and legal standards governing the use of human genomic data. Access is restricted to qualified researchers who have been approved to access and use the data for legitimate research purposes. Data requests should be directed to the Dr. Paul Bunn (paul.bunn@ucdenver.edu) for Lung Cancer Mutation Consortium. Typically, a response will be received within one month. The OPMD dataset is subject to controlled access to ensure the responsible and ethical use of this sensitive data. Access is restricted to qualified researchers who have been approved to access and use the data for legitimate research purposes. To request access to the OPMD dataset, interested researchers should contact Dr. Vassiliki Papadimitrakopoulou (vali.papa@pfizer.com) with a formal request. Typically, a response will be received within one month. Source data are provided with this paper.

## Code availability

Scripts for Ceograph model is available at https://github.com/sdw95927/Ceograph/.

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

## Acknowledgements

This study is partially supported by National Institutes of Health (P50CA070907, R01GM140012, R01DE030656, R01GM115473, 1U01CA249245, 1U01AI169298, R35GM136375, P30CA008748 and P30CA142543) and the Cancer Prevention and Research Institute of Texas (CPRIT RP180805 and CPRIT RP230330). The authors thank Jessie Norris for helping us to edit this manuscript.

## Author contributions

S.W., Y.X. and G.X. designed the study. S.W. and R.R. implemented and trained the neural network model. Q.Z. checked the code and data. S.W., R.R., Q.Z., D.M.Y., C.J.W., Y.X. and G.X. analyzed the data and wrote the manuscript. J.B. and Z.C. provided critical inputs as pathologists. J.M. and M.G.K. provided the LCMC data sets and supervised the study. C.R.P. provided OPMD dataset. X.Z., X.Z., C.R.W., S.Z., C.R.P., M.G.K., J.M., Y.X. provided critical inputs. All co-authors read and commented on the manuscript.

## Competing interests

The authors declare no competing interests.
