## [Peer Review File · Nature Communications]

Deep Learning of Cell Spatial Organizations Identifies Clinically Relevant Insights in Tissue ImagesREVIEWER COMMENTS

Reviewer #1 (Remarks to the Author):

The authors present Ceograph, a cell spatial organization-based graph convolutional neural network which takes graph representations of pathology images and can be used to predict histology variables and clinical outcomes. Overall, this work is an important demonstration of how encoded biological knowledge can improve performance and interpretability of predictive models by directing algorithms to discover connections between biologically meaningful phenomena, rather than expecting to distinguish between signal and noise automatically. The graph representations of pathology images take up less memory and result in fewer parameters, likely leading to fast model training compared to models trained on raw images which justifies a pre-training step of extracting graph representations from the images themselves.

The noteworthy results are organised into three use cases: (1) detecting lung cancer subtype, (2) predicting malignant transformation of oral potentially malignant disorders and (3) predicting treatment response in lung cancer. In the first use case, Ceograph correctly categorises slides as ADC or SCC with very high accuracy. This result is impressive but perhaps not surprising since the histological differences in cell morphology between ADC and SCC were used as features in the model. However, the result is important and meaningful, demonstrating the benefits of bringing together biological knowledge and traditionally 'black box' models to uncover new insights and improve predictions. The key result of the second use case is that the Ceograph model can detect known pathological features which indicate patients with higher risk of malignancy. In the third use case, the authors illustrate that the model can be trained on pre-treatment pathology slides to predict whether patients will benefit or not from EGFR targeted therapy.

I invite the authors to consider the following comments which may improve the quality of the manuscript.

- It would be good to have one table indicating which data were used to answer which research question (most importantly the training, testing and validation set sizes).
- The references to figures jump around a lot between the main figures and the supplementary and are not always referred to in order. I think some of the material in the supplementary is important enough to be included in the main text. Would it be possible to improve the flow by including more figures in the main text, or more panels in the existing figures?
- For the second research question (malignant transformation prediction of OPMD), the authors explain that datapoints were classified as 'low risk CFS > 68.5 months' and 'high risk CFS < 42.3 months'. It isn't clear to me how data from patients with $42.3m < CFS < 68.5m$ are categorised.

- The authors claim that the use case of using Ceograph to predict response to treatment could enhance treatment efficacy by allowing clinicians to bypass ineffective treatments. I find this a very ambitious claim!

Some of the p-values reported are extremely small (10^{-41} , 10^{-34}). I invite the authors to double check these since visually from the supplementary figures the differences between the groups does not seem very large. If these numbers turn out to be correct due to the large sample sizes involved, I propose that the authors consider replacing with 'p<0.001' or similar.

- In the final sentence of the penultimate paragraph of the introduction, the authors claim that ceograph can identify features with "driving effects" but I am not sure that this is what they have shown. Similarly in the discussion, the authors claim that the models lead to "biological insights into how cells assemble and interact to produce diverse functional consequences". I rather understand that Ceograph is able to detect correlation between image/graph features and outcomes, but further investigation would be required to understand if there is causation involved.

- I would be interested to understand how graphs were derived for the cells on the edges of the ROIs which would artificially appear to have fewer neighbours.

- The splits between training, validation and testing groups are not very clearly explained and sometimes the testing groups seem rather small. I would be interested to understand why this is.

- The authors claim that Ceograph "outperformed all other CNN-based deep learning models" and I think it would be reasonable to clarify that Ceograph outperformed the other CNN-based deep learning models that were investigated as part of this work.

- In the section "Ceograph predicts risk of malignant transformation of OPMD", the authors claim that the reduction in AUC between 24 months and 50 months is 'modest', but I would suggest that the difference between AUC=0.915 and AUC=0.797 is not a modest reduction but quite a notable one.

Reviewer #2 (Remarks to the Author):

In this paper, the authors introduced a cell spatial organization-based graph convolutional network designed to analyze cell spatial organization in pathology images called Ceograph. Authors tested this approach in the context of predicting lung cancer subtyping (adeno vs squamous cell carcinoma), prognosis in oral potentially malignant disorders, and prediction to TKI targeted therapy in patients with metastatic lung cancer having EGFR mutation. They also provide some insights on the interpretation of the results.

In general, the paper is well written and easy to follow. The methodology seems reproducible and some of the results look promising. However, there are important points that need to be clarified, adjusted, or further discussed to make the manuscript suitable for publication.

- An important concern is regarding the prognostic analysis. Authors arbitrarily defined that low-risk patients were those with cancer-free survival (CFS) > 68.5 months, and high-risk patients were those with CFS ≤ 42.3 months. How was this threshold set? What happened with the patients in the middle (> 42.3 and < 68.5)? More importantly, authors did not clarify what happened with censored patients. From the explanation, it looks like a patient who was censored before 42.3 months could have been classified as “high risk,” which is wrong as there is no certainty whether or not the patient had cancer. For prognostic models, strict binary classification is not recommended since it does not properly model time and censored patients. In these cases, the recommendation is using a neural network coupled with a loss function specific to survival analysis (see for example [10.1073/pnas.1717139115](https://doi.org/10.1073/pnas.1717139115))
- The previous comment also applies for the predictive analysis. Authors defined arbitrarily a threshold of 31 months for “response to treatment”, why? What happened to the censored patients? Finally, the multivariable survival analysis appears not to be statistically significant for their predictions (benefitting vs non-benefitting).
- Authors compared their approach with ResNet101; however, comparing with a Graph Neural Network (GNN) seems like a fairer option.
- A better explanation of the method by Simonovsky et al., in which Ceograph relies heavily on, is desirable. It is not completely clear how the data is input to the network.
- Authors employed a state-of-the-art method for nuclear identification and classification, HD-Staining. However, they did not mention any quality check of this method for the datasets of this paper. Did a pathologist review and validated predictions made by HD-Staining to make sure they were good enough?

Minor:

- The authors should tone down some sections of the paper where they claim that there is a need for computational methods capable of characterizing complex cell spatial organization. In fact, they claim to be the first paper to characterize cell spatial organization using graph models at single cell level. Although this could be true in the deep learning arena, there are multiple handcrafted approaches that have done so, see for example: [10.1016/j.media.2020.101696](https://doi.org/10.1016/j.media.2020.101696), [10.1016/j.media.2020.101903](https://doi.org/10.1016/j.media.2020.101903) , [10.1038/s41698-022-00277-5](https://doi.org/10.1038/s41698-022-00277-5).
- Results of the research work should not be described in the last paragraph of the Introduction

The reviewers' comments verbatim are in blue. Our responses are in plain black text. Our corresponding changes were highlighted in red in the manuscript.

REVIEWER COMMENTS

Reviewer #1 (Remarks to the Author):

The authors present Ceograph, a cell spatial organization-based graph convolutional neural network which takes graph representations of pathology images and can be used to predict histology variables and clinical outcomes. Overall, this work is an important demonstration of how encoded biological knowledge can improve performance and interpretability of predictive models by directing algorithms to discover connections between biologically meaningful phenomena, rather than expecting to distinguish between signal and noise automatically. The graph representations of pathology images take up less memory and result in fewer parameters, likely leading to fast model training compared to models trained on raw images which justifies a pre-training step of extracting graph representations from the images themselves.

The noteworthy results are organised into three use cases: (1) detecting lung cancer subtype, (2) predicting malignant transformation of oral potentially malignant disorders and (3) predicting treatment response in lung cancer. In the first use case, Ceograph correctly categorises slides as ADC or SCC with very high accuracy. This result is impressive but perhaps not surprising since the histological differences in cell morphology between ADC and SCC were used as features in the model. However, the result is important and meaningful, demonstrating the benefits of bringing together biological knowledge and traditionally 'black box' models to uncover new insights and improve predictions. The key result of the second use case is that the Ceograph model can detect known pathological features which indicate patients with higher risk of malignancy. In the third use case, the authors illustrate that the model can be trained on pre-treatment pathology slides to predict whether patients will benefit or not from EGFR targeted therapy.

I invite the authors to consider the following comments which may improve the quality of the manuscript.

- It would be good to have one table indicating which data were used to answer which research question (most importantly the training, testing and validation set sizes).

Response: We are grateful for the positive comments and constructive suggestions. As recommended, we have created Supplemental Table 1 (See below) to summarize the sample size for each dataset for each research question.

Supplemental Table 1. A summary of datasets for each research question

Research Question	Model development	External Validation
Research Question 1: Classification for lung adenocarcinoma (ADC) vs. squamous cell carcinoma (SCC)	TCGA LUAD dataset (Pathology slides: n=469, Patients: n=422) and LUSC dataset (Pathology slides: n=379, Patients: n=379)	NLST dataset (Pathology slides: n=496, Patients: n=286)
Research Question 2: Risk prediction of malignant transformation in Oral potentially malignant disorders (OPMD)	OPMD 1 dataset (Pathology slides: n=23, Patients: n=23)	OPMD 2 dataset (Pathology slides: n=53, Patients: n=53)
Research Question 3: EGFR TK1 Ttx response prediction	LCMC 1 dataset (Pathology slides: n=115, Patients: n=98)	LCMC 1 dataset (Pathology slides: n=137, Patients: n=126)

- The references to figures jump around a lot between the main figures and the supplementary and are not always referred to in order. I think some of the material in the supplementary is important enough to be included in the main text. Would it be possible to improve the flow by including more figures in the main text, or more panels in the existing figures?

Response: We appreciate the thoughtful feedback regarding the organization of figures between the main text and supplementary material. In response to reviewer's valuable suggestions, we have made the following revisions to the manuscript:

- The original Supplementary Figure 1B has been incorporated into the revised Figure 1.
- The original Supplementary Figure 2 is now included in the revised Figure 3.
- For a more coherent comparison, we have combined the original Supplementary Figure 4 C, Supplementary Figure 5 A&B, and Supplementary Figure 6 A&B into a single figure. This will facilitate a clearer head-to-head comparison of results from the TCGA testing dataset and the NLST dataset using different methodologies.

We hope these revisions will enhance the flow of information and provide readers with a more streamlined narrative.

- For the second research question (malignant transformation prediction of OPMD), the authors explain that data points were classified as 'low risk CFS>68.5 months' and 'high risk CFS <42.3 months'. It isn't clear to me how data from patients with 42.3m<CFS<68.5m are categorised.

Response: We thank the reviewer for highlighting this point and allowing us to clarify our methodology. Given the limited sample availability in OPMD malignancy transformation research, our model's training/validation strategy was to utilize a smaller set of cases with distinct clinical phenotypes for training, while employing a larger, more clinically representative independent cohort for validation. For the training data, the outcome was designated as a binary variable—either high risk or low risk. Using the Ceograph-based classification model, we aimed to categorize new cases into these risk categories. We then assessed the prediction performance of the model in the independent cohort by comparing the predicted risk groups (high vs. low) to the observed cancer-free survival time using KM curves and the log-rank test.

In line with this strategy, Dr. Curtis Pickering, an OPMD expert from MD Anderson and co-author of this manuscript, handpicked 17 patients with a follow-up time exceeding 68.5 months (who did not develop oral cancer) as low-risk cases. Conversely, 6 patients who developed oral cancer within 42.3 months were defined as high risk. These selections from the MD Anderson Cancer Center's clinical cases, which included high-quality pathology slides and clinical annotations, constituted the training set, named the OPMD1 dataset. Notably, in the OPMD1 dataset, no patients had CFS values ranging between 42.3 months and 68.5 months.

For the test set, termed the OPMD2 dataset, we considered data from patients in the placebo arm (n=75) of the Erlotinib Prevention of Oral Cancer (EPOC) trial at MD Anderson Cancer Center. After excluding patients with potential confounding comorbidities, 53 patients remained. This group encompassed patients with varying CFS values and censoring/progression statuses.

To simplify our model training, we exclusively selected patients who developed cancer in a relatively short timeframe as the high-risk group. Those who didn't develop cancer during an extended follow-up were classified as low risk. This method's inherent limitation is its restricted capacity to utilize the information from censored patients. As another reviewer pointed out, an alternative approach could involve using time-to-event as outcome variables, coupled with a neural network paired with a survival-specific loss function. Such a method might bolster predictive accuracy. We have elaborated on this further in a new paragraph in the Discussion Section.

In the revised manuscript, we clarified the definition of the high- and low-risk group as below:

In the Discussion Section:

“In the process of training our model, we employed a deliberate stratification approach: patients who developed cancer quickly were categorized as the high-risk group, while those who remained cancer-free over an extended follow-up period were designated as the low-risk group. This strategy aimed to reduce ambiguity in survival outcomes during model training. However, this method has a notable limitation: it does not fully utilize on the information from patients with censored outcomes. As highlighted by recent studies^[39], coupling a neural network with a loss function specifically designed for survival analysis could potentially improve predictive performance for survival outcomes.”

[Ref 39]. Mobadersany P, Yousefi S, Amgad M, Gutman DA, Barnholtz-Sloan JS, Velázquez Vega JE, Brat DJ, Cooper LAD. Predicting cancer outcomes from histology and genomics using convolutional networks. *Proceedings of the National Academy of Sciences*. 2018;115(13):E2970-E9. doi: doi:10.1073/pnas.1717139115.

In the Dataset subsection of Method Section:

OPMD1 and OPMD2 datasets for malignant transformation prediction

Patients with baseline OPMD were biopsied and monitored at the Department of Head and Neck Surgery at The University of Texas MD Anderson Cancer Center (MDACC). The OPMD pathology images, reviewed by expert pathologists and scanned at 40X magnification, were acquired in two distinct batches. These batches are referred to as the OPMD1 and OPMD2 datasets, respectively. The risk of malignant transformation was quantified using the Cancer-free Survival (CFS) time, defined as the duration from the diagnosis of OPMD to the diagnosis of oral cancer or the last follow-up. Patients with concurrent cancers were excluded from the OPMD datasets.

Aligning with our modeling strategy and the set exclusion criteria, we identified 17 patients who had a follow-up duration exceeding 68.5 months without developing oral cancer (CFS>68.5 months) as low-risk patients. Conversely, 6 patients who developed oral cancer within 42.3 months (CFS<42.3 months and event = true) were classified as high-risk and were included in the OPMD1 training data. For validation, we used the OPMD2 dataset,

we used data from 53 patients in the Erlotinib Prevention of Oral Cancer (EPOC) trial at MDACC. This group encompasses patients with a range of CFS durations and varying censoring/progression statuses. The most extended follow-up duration recorded was 126 months.

In the Ceograph For Malignant Transformation Prediction subsection of Method Section:

Given the limited sample availability for OPMD malignant transformation research, our modeling strategy relied on a select set of cases with clear clinical phenotypes as the training dataset (OPMD 1 data) and a larger, independent cohort (OPMD 2 data) mirroring real-world clinical scenarios for validation. The training data utilized a binary outcome—categorized as high-risk or low-risk. A Ceograph-based classification model was developed to predict a new case as a high risk or low risk case. The prediction performance was then validated in the independent cohort by examining the association between predicted risk groups and the observed time to cancer using K-M curves and log-rank tests. Cancer-free survival (CFS) time and event status (progress to oral cancer or not) were used to quantify risk of malignant transformation and defined as time from diagnosis of OPMD to diagnosis of oral cancer, or last follow-up.

Patients at MDACC with OPMD were biopsied and monitored. Within the OPMD1 dataset, 17 patients who did not develop cancer over a span of 68.5 months were tagged as low-risk, while 6 patients who exhibited cancer signs within 42.3 months were considered high-risk. The OPMD2 dataset, derived from 53 patients involved in the Erlotinib Prevention of Oral Cancer trial, was designated for validation, comprising patients with diverse CFS durations (refer to the Dataset Section). During each training epoch, a unique 2,048×2,048 pixel image patch containing a minimum of 50 epithelial nuclei was randomly chosen from each patient. Hence, image patches from the same patient varied across epochs. For the testing set, 100 distinct 2,048×2,048 pixel image patches were randomly selected from each OPMD2 dataset patient, ensuring each patch had at least 50 epithelial nuclei. Image patches from both OPMD1 and OPMD2 were each transformed into individual graphs.

In the Results Section:

We utilized the morphological characteristics and spatial organization of the four major nuclei types within skin tissue: stratum corneum, stratum basale, other strata, and non-epithelium. Subsequently, we trained a Ceograph based OPMD malignant transformation risk prediction model using the OPMD1 dataset (refer to the Method Section for a detailed description of the OPMD1 and OPMD2 datasets) to distinguish between high-risk and low-risk groups (**Supplemental Figure 6**). In the OPMD2 independent testing set, the predicted high-risk group demonstrated significantly shorter time to malignant transformation (defined as cancer-free survival time, CFS) compared to the low-risk group (**Figure 4B**, $p = 0.012$; high- vs. low-risk, Hazard Ratio [HR] = 3.17, 95% Confidence Interval [CI] 1.22 – 3.22).

- The authors claim that the use case of using Ceograph to predict response to treatment could enhance treatment efficacy by allowing clinicians to bypass ineffective treatments. I find this a very ambitious claim!

Response: We thank the reviewer for this comment and agree with the reviewer that the original claim is over ambitious. We have rephrased this paragraph as below:

“Developing predictive models to predict patients' response to treatment before actual administration is of great clinical importance. This might help improve treatment outcomes and potentially reduce side effects. It could also offer clinicians insights to potentially avoid less effective treatments.”

Some of the p-values reported are extremely small (10^{-41} , 10^{-34}). I invite the authors to double check these since visually from the supplementary figures the differences between the groups does not seem very large. If these numbers turn out to be correct due to the large sample sizes involved, I propose that the authors consider replacing with ' $p < 0.001$ ' or similar.

Response: We thank the reviewer for this comment. We double-checked the numbers and the p values are correct. These extremely small p values are due to the large sample size, as these comparisons are between groups with very large numbers of cells (more than 15,000 cells). We have replaced the extremely small p values with ' $p < 0.001$ ', as suggested by the reviewer.

- In the final sentence of the penultimate paragraph of the introduction, the authors claim that Ceograph can identify features with "driving effects" but I am not sure that this is what they have shown. Similarly in the discussion, the authors claim that the models lead to "biological insights into how cells assemble and interact to produce diverse functional consequences". I rather understand that Ceograph is able to detect correlation between image/graph features and outcomes, but further investigation would be required to understand if there is causation involved.

Response: We thank the reviewer for the insightful comment. We agree with the reviewer on the distinction between correlation and causation. When we mentioned that Ceograph can identify features with "driving effects," our intention was to highlight its capability in pinpointing salient features of interest. However, we acknowledge that establishing causation requires a more in-depth analysis and investigation. We have rephrased the paragraphs in Introduction and Discussion as below:

In Introduction Section

"By assessing the relationship between individual cell morphology and interactions with diagnosis and clinical outcomes, Ceograph highlights image features that correlate with potential biological significance. This offers a starting point for further exploration into biological and clinical interpretations."

In Discussion Section

"The results suggest that Ceograph models offer interpretability, indicating potential correlations in how cells assemble and interact across various contexts. This provides avenues for further exploration into their functional implications."

- I would be interested to understand how graphs were derived for the cells on the edges of the ROIs which would artificially appear to have fewer neighbours.

Response: We thank the reviewer for raising the insightful concern about the edge effect in cells located on the boundaries of our regions of interest (ROIs). The Ceograph method constructs a directed graph for each image patch using k-Nearest Neighbors, which is based on Euclidean distance, with the direction pointing from a cell to its neighbors (as depicted in Figure 1A). We set k to 8 to encompass the immediate neighbors of each nucleus.

As a consequence, cells on the edge of the ROI are conditioned to have all their neighbors within the ROI. This setup gives rise to the edge effect, which could potentially lead to certain biases. This is an intrinsic limitation of constructing a graph from a ROI. However, considering the substantial size of the ROIs, which houses thousands of cells, the magnitude of the edge effect is expected to be relatively minimal.

In response to this observation, we have added a paragraph in the Discussion Section (copied below) that elaborates on this methodological limitation. We sincerely appreciate the reviewer's insightful feedback and the opportunity to improve the clarity and thoroughness of our work.

In Discussion Section

“The edge effect is a known challenge in image analysis that focuses on regions of interest (ROIs), particularly with our Ceograph method that utilizes k-Nearest Neighbors for graph construction. When cells lie on the boundaries of ROIs, they're constrained to have neighbors only within that ROI. This design can introduce biases for these boundary cells. However, considering the substantial size of our typical ROIs containing a large number of cells, while the number of cells affected by the edge effect is relatively small. It is important to recognize this inherent limitation and consider it in future methodological refinements.”

- The splits between training, validation and testing groups are not very clearly explained and sometimes the testing groups seem rather small. I would be interested to understand why this is.

Response: We thank the reviewer for this comment and the opportunity to improve the clarity of our work. For each research question, a Ceograph model was developed in one dataset and independently validated on another dataset:

- (1) For the classification for lung adenocarcinoma (ADC) vs. squamous cell carcinoma (SCC), the Ceograph model was developed from TCGA LUAD dataset (n=469) and LUSC dataset (n=379) and validated in the NLST dataset (n= 496). Since this is the first Ceograph model we developed, we split the TCGA LUAD + LUSC with 848 patients (469 + 379) in total into internal training, validation and testing groups with (70%, 10% and 20% of patients in each group). This model was then externally validated in the NLST dataset (n= 496)
- (2) For the risk prediction of malignant transformation in Oral potentially malignant disorders (OPMD), the model was trained on OPMD 1 dataset (n=23) and validated on the OPMD 2 dataset (n=53). Due to the limited samples availability for OPMD malignance transformation research, our model training/validation strategy was to use a small set of cases with the most distinguished clinical phenotypes as the training dataset and another larger independent cohort similar to real-world clinical cases as the validation dataset. Following this modeling strategy, Dr. Curtis Pickering (an OPMD expert from MD Anderson and a co-author in this manuscript) selected 17 patients with follow up time longer than 68.5 months without developing oral cancer (CFS>68.5 months) as low risk patients, and 6 patients who developed oral cancer within 42.3 months (CFS <42.3 months) as high risk patients from MDACC clinical cases with high quality pathology slides and clinical annotation as the training set (OPMD1 dataset) for the model training purpose. In the test set (OPMD2 dataset), we used the data from patients in the placebo arm (n=75) of the Erlotinib Prevention of Oral Cancer (EPOC) trial at MD Anderson Cancer Center. After excluding patients with potential cofounding comorbidities, there are 53 patients remaining, which include patients with various CFS and censoring/progression status.

(3) For EGFR TK1 Ttx response prediction, the model was trained on LCMC 1 dataset (n=98) and validated on the LCMC 2 dataset (n=126). The LCMC 1 cohort was derived from patients who enrolled in the multi-institutional clinical trial established to study different oncogenic drivers and corresponding target therapies between 2009 and 2012 ^[40], while the LCMC 2 cohort was derived from patients enrolled in the multi-institutional LCMC2 trial between January 1, 2013 and December 1, 2015 ^[41]

[Ref 40] Sholl LM, Aisner DL, Varella-Garcia M, Berry LD, Dias-Santagata D, Wistuba, II, Chen H, Fujimoto J, Kugler K, Franklin WA, Iafrate AJ, Ladanyi M, Kris MG, Johnson BE, Bunn PA, Minna JD, Kwiatkowski DJ, Investigators L. Multi-institutional Oncogenic Driver Mutation Analysis in Lung Adenocarcinoma: The Lung Cancer Mutation Consortium Experience. J Thorac Oncol. 2015;10(5):768-77. Epub 2015/03/05. PubMed PMID: 25738220; PMCID: PMC4410843.

[Ref 41] Aisner DL, Sholl LM, Berry LD, Rossi MR, Chen H, Fujimoto J, Moreira AL, Ramalingam SS, Villaruz LC, Otterson GA, Haura E, Politi K, Glisson B, Cetnar J, Garon EB, Schiller J, Waqar SN, Sequist LV, Brahmer J, Shyr Y, Kugler K, Wistuba, II, Johnson BE, Minna JD, Kris MG, Bunn PA, Kwiatkowski DJ, investigators L. The Impact of Smoking and TP53 Mutations in Lung Adenocarcinoma Patients with Targetable Mutations-The Lung Cancer Mutation Consortium (LCMC2). Clin Cancer Res. 2018;24(5):1038-47. Epub 2017/12/09. PubMed PMID: 29217530.

In the revised manuscript, we added **Supplemental Table 1** to summarize the training, testing and validation data for each research question.

Supplemental Table 1. A summary of datasets for each research question

Research Question	Model development	External Validation
Research Question 1: Classification for lung adenocarcinoma (ADC) vs. squamous cell carcinoma (SCC)	TCGA LUAD dataset (Pathology slides: n=469, Patients: n=422) and LUSC dataset (Pathology slides: n=379, Patients: n=379)	NLST dataset (Pathology slides: n=496, Patients: n=286)
Research Question 2: Risk prediction of malignant transformation in Oral potentially malignant disorders (OPMD)	OPMD 1 dataset (Pathology slides: n=23, Patients: n=23)	OPMD 2 dataset (Pathology slides: n=53, Patients: n=53)
Research Question 3: EGFR TK1 Ttx response prediction	LCMC 1 dataset (Pathology slides: n=115, Patients: n=98)	LCMC 1 dataset (Pathology slides: n=137, Patients: n=126)

- The authors claim that Ceograph “outperformed all other CNN-based deep learning models” and I think it would be reasonable to clarify that Ceograph outperformed the other CNN-based deep learning models that were investigated as part of this work.

Response: We agree with the reviewer on this comment. In the revised manuscript, we modified the claim as suggested by the reviewer. We copy the paragraph below:

“In the lung cancer pathology subtype classification task, Ceograph outperformed **the other CNN-based deep learning models that were investigated as part of this work** and traditional feature-based machine learning methods (**Table 2**).”

- In the section “Ceograph predicts risk of malignant transformation of OPMD”, the authors claim that the reduction in AUC between 24 months and 50 months is ‘modest’, but I would suggest that the difference between AUC=0.915 and AUC=0.797 is not a modest reduction but quite a notable one.

Response: We agree with the reviewer on this comment. In the revised manuscript, we modified the claim as suggested by the reviewer. We copy the paragraph below:

The reduction in AUC for the 50-month prediction, compared to the 24-month prediction, may be attributed to the accumulation of additional confounding factors over the longer prediction timeline.

Reviewer #2 (Remarks to the Author):

In this paper, the authors introduced a cell spatial organization-based graph convolutional network designed to analyze cell spatial organization in pathology images called Ceograph. Authors tested this approach in the context of predicting lung cancer subtyping (adeno vs squamous cell carcinoma), prognosis in oral potentially malignant disorders, and prediction to TKI targeted therapy in patients with metastatic lung cancer having EGFR mutation. They also provide some insights on the interpretation of the results.

In general, the paper is well written and easy to follow. The methodology seems reproducible and some of the results look promising. However, there are important points that need to be clarified, adjusted, or further discussed to make the manuscript suitable for publication.

- An important concern is regarding the prognostic analysis. Authors arbitrarily defined that low-risk patients were those with cancer-free survival (CFS) > 68.5 months, and high-risk patients were those with CFS ≤ 42.3 months. How was this threshold set? What happened with the patients in the middle (> 42.3 and < 68.5)? More importantly, authors did not clarify what happened with censored patients. From the explanation, it looks like a patient who was censored before 42.3 months could have been classified as “high risk,” which is wrong as there is no certainty whether or not the patient had cancer. For prognostic models, strict binary classification is not recommended since it does not properly model time and censored patients. In these cases, the recommendation is using a neural network coupled with a loss function specific to survival analysis (see for example [10.1073/pnas.1717139115](https://doi.org/10.1073/pnas.1717139115))

Response: We thank the reviewer for highlighting this point and allowing us to clarify our methodology. Given the limited sample availability in OPMD malignancy transformation research, our model's training/validation strategy was to utilize a smaller set of cases with distinct clinical phenotypes for training, while employing a larger, more clinically representative independent cohort for validation. For the training data, the outcome was designated as a binary variable—either high risk or low risk. Using the Ceograph-based classification model, we aimed to categorize new cases into these risk categories. We then assessed the prediction performance of the model in the independent cohort by comparing the predicted risk groups (high vs. low) to the observed cancer-free survival time using KM curves and the log-rank test.

In line with this strategy, Dr. Curtis Pickering, an OPMD expert from MD Anderson and co-author of this manuscript, handpicked 17 patients with a follow-up time exceeding 68.5 months (who did not develop oral cancer) as low-risk cases. Conversely, 6 patients who developed oral cancer within 42.3 months were defined as high risk. These selections from the MD Anderson Cancer Center's clinical cases, which included high-quality pathology slides and clinical annotations, constituted the training set, named the OPMD1 dataset. Notably, in the OPMD1 dataset, no patients had CFS values ranging between 42.3 months and 68.5 months.

For the test set, termed the OPMD2 dataset, we considered data from patients in the placebo arm (n=75) of the Erlotinib Prevention of Oral Cancer (EPOC) trial at MD Anderson Cancer Center. After excluding patients with

potential confounding comorbidities, 53 patients remained. This group encompassed patients with varying CFS values and censoring/progression statuses.

The original terminology “prognostic model’ was a bit misleading. Although the purpose of the analysis was to predict the prognosis of OPMD, the actual Celgraph model is a classification model (with binary outcomes). The predicted risk groups and predicted risk scores (probability of developing cancer) were associated with time to cancer in the validation data. To reduce the confusion, we changed it from the prognostic model to OPMD malignant transformation risk prediction model.

To simplify our model training, we only selected patients who developed cancer in a relatively short timeframe as the high-risk group, and those who didn't develop cancer during an extended follow-up were classified as low risk. This method's inherent limitation is its restricted capacity to utilize the information from censored patients. As another reviewer pointed out, an alternative approach could involve using time-to-event as outcome variables, coupled with a neural network paired with a survival-specific loss function. Such a method might bolster predictive accuracy. We have elaborated on this further in a new paragraph in the Discussion Section.

In the Discussion Section:

“In the process of training our model, we employed a deliberate stratification approach: patients who developed cancer quickly were categorized as the high-risk group, while those who remained cancer-free over an extended follow-up period were designated as the low-risk group. This strategy aimed to reduce ambiguity in survival outcomes during model training. However, this method has a notable limitation: it does not fully utilize on the information from patients with censored outcomes. As highlighted by recent studies^[39], coupling a neural network with a loss function specifically designed for survival analysis could potentially improve predictive performance for survival outcomes.”

[Ref 39]. Mobadersany P, Yousefi S, Amgad M, Gutman DA, Barnholtz-Sloan JS, Velázquez Vega JE, Brat DJ, Cooper LAD. Predicting cancer outcomes from histology and genomics using convolutional networks. *Proceedings of the National Academy of Sciences*. 2018;115(13):E2970-E9. doi: doi:10.1073/pnas.1717139115.

In the revised manuscript, we clarified the definition of the high- and low-risk group as below:

In the Dataset subsection of Method Section:

OPMD1 and OPMD2 datasets for malignant transformation prediction

Patients with baseline OPMD were biopsied and monitored at the Department of Head and Neck Surgery at The University of Texas MD Anderson Cancer Center (MDACC). The OPMD pathology images, reviewed by expert pathologists and scanned at 40X magnification, were acquired in two distinct batches. These batches are referred to as the OPMD1 and OPMD2 datasets, respectively. The risk of malignant transformation was quantified using the Cancer-free Survival (CFS) time, defined as the duration from the diagnosis of OPMD to the diagnosis of oral cancer or the last follow-up. Patients with concurrent cancers were excluded from the OPMD datasets.

Aligning with our modeling strategy and the set exclusion criteria, we identified 17 patients who had a follow-up duration exceeding 68.5 months without developing oral cancer (CFS>68.5 months) as low-risk patients. Conversely, 6 patients who developed oral cancer within 42.3 months (CFS<42.3 months and event = true) were classified as high-risk and were included in the OPMD1 training data. For validation, we used the OPMD2 dataset, we used data from 53 patients in the Erlotinib Prevention of Oral Cancer (EPOC) trial at MDACC. This group

encompasses patients with a range of CFS durations and varying censoring/progression statuses. The most extended follow-up duration recorded was 126 months.

In the Ceograph For Malignant Transformation Prediction subsection of Method Section:

Given the limited sample availability for OPMD malignant transformation research, our modeling strategy relied on a select set of cases with clear clinical phenotypes as the training dataset (OPMD 1 data) and a larger, independent cohort (OPMD 2 data) mirroring real-world clinical scenarios for validation. The training data utilized a binary outcome—categorized as high-risk or low-risk. A Ceograph-based classification model was developed to predict a new case as a high risk or low risk case. The prediction performance was then validated in the independent cohort by examining the association between predicted risk groups and the observed time to cancer using K-M curves and log-rank tests. Cancer-free survival (CFS) time and event status (progress to oral cancer or not) were used to quantify risk of malignant transformation and defined as time from diagnosis of OPMD to diagnosis of oral cancer, or last follow-up.

Patients at MDACC with OPMD were biopsied and monitored. Within the OPMD1 dataset, 17 patients who did not develop cancer over a span of 68.5 months were tagged as low-risk, while 6 patients who exhibited cancer signs within 42.3 months were considered high-risk. The OPMD2 dataset, derived from 53 patients involved in the Erlotinib Prevention of Oral Cancer trial, was designated for validation, comprising patients with diverse CFS durations (refer to the Dataset Section). During each training epoch, a unique 2,048×2,048 pixel image patch containing a minimum of 50 epithelial nuclei was randomly chosen from each patient. Hence, image patches from the same patient varied across epochs. For the testing set, 100 distinct 2,048×2,048 pixel image patches were randomly selected from each OPMD2 dataset patient, ensuring each patch had at least 50 epithelial nuclei. Image patches from both OPMD1 and OPMD2 were each transformed into individual graphs.

In the Results Section:

We utilized the morphological characteristics and spatial organization of the four major nuclei types within skin tissue: stratum corneum, stratum basale, other strata, and non-epithelium. Subsequently, we trained a Ceograph based OPMD malignant transformation risk prediction model using the OPMD1 dataset (refer to the Method Section for a detailed description of the OPMD1 and OPMD2 datasets) to distinguish between high-risk and low-risk groups (**Supplemental Figure 6**). In the OPMD2 independent testing set, the predicted high-risk group demonstrated significantly shorter time to malignant transformation (defined as cancer-free survival time, CFS) compared to the low-risk group (**Figure 4B**, $p = 0.012$; high- vs. low-risk, Hazard Ratio [HR] = 3.17, 95% Confidence Interval [CI] 1.22 – 3.22).

- The previous comment also applies for the predictive analysis. Authors defined arbitrarily a threshold of 31 months for “response to treatment”, why? What happened to the censored patients?

Response: We are grateful for the reviewer's feedback and aim to elucidate our method further. In the LCMC1 training dataset, the clinical outcomes were dichotomized into 'benefitting' and 'non-benefitting' groups using an overall survival (OS) cutoff of 31 months (which is very close to the median OS time at 31.2 months). In this LCMC1 dataset, the first censoring event happened at a follow-up period of 31.2 months, as depicted in the Kaplan-Meier survival curve provided below. Consequently, all patients in this cohort who had a follow-up duration shorter than 31 months (OS < 31 months) were dead (event=true) and categorized as non-benefitting

cases. Conversely, patients who survived beyond 31 months (OS > 31 months) were classified as benefitting cases. Subsequently, we developed a Ceograph-based classification model to prognosticate whether a new case would belong to the benefitting or non-benefitting category. The model's predictive accuracy was later assessed in an independent cohort (LCMC2) by examining the relationship between the forecasted benefit groups and the actual survival duration.

In the revised manuscript, we clarified the definition of the non-benefitting and benefitting group as below:

In the Method Section:

“The predictive Ceograph model was trained on the LCMC1 dataset and independently tested on the LCMC2 dataset. OS, defined as time from diagnosis of metastatic disease to death or last follow-up, was used as the outcome. Within the LCMC1 dataset, patients who died within 31 months (OS ≤ 31 months and event=true) were categorized as non-benefitting, while those who survived longer than 31 months (OS > 31 months) were classified as benefitting (Supplemental Table 1). The OS cutoff was selected to ensure non-benefitting and benefitting groups are relatively balanced.”

Finally, the multivariable survival analysis appears not to be statistically significant for their predictions (benefitting vs. non-benefitting).

Response: In the testing cohort (LCMC2 dataset), there are four groups of patients separated by two factors, the predicted benefitting group (benefitting vs. non-benefitting groups) and EGFR targeted therapy (with vs. without) summarized in Table below:

	with EGFR targeted therapy	without EGFR targeted therapy
Predicted benefitting groups	45 patients (green line)	14 patients (red line)
Predicted non-benefitting groups	44 patients (purple line)	19 patients (blue line)

In the multivariate survival analysis for a predictive signature or biomarker, the efficacy of the signature to predict treatment response is usually measured by the interaction between the predicted response group and the treatment group. In our results from LCMC2 testing cohort, the term "Interaction between EGFR Tx and benefitting" in Figure 5C (p-value=0.0027) is statistically significant, suggesting that the model is predictive of prolonged survival time in patients with EGFR mutations who received EGFR TKI treatment. In Figure 5C, the

benefitting vs. non-benefitting results (HR=2.72 and p=0.11) refer to survival differences between benefitting and non-benefitting groups among patients who did not receive EGFR Ttx (red vs. blue lines in Figure 5B).

- Authors compared their approach with ResNet101; however, comparing with a Graph Neural Network (GNN) seems like a fairer option.

Response: Thank you for your constructive feedback. We acknowledge the reviewer's point that comparing our method with a Graph Neural Network (GNN) would provide a more direct and relevant benchmark, given the graph-based nature of our approach. However, to the best of our knowledge, there is no GNN network developed to incorporate cell spatial organization from tissue images. As a result, we chose to compare with ResNet101 to demonstrate the advantages of our method against a traditional deep learning architecture, which is commonly used for image-based tasks.

- A better explanation of the method by Simonovsky et al., in which Ceograph relies heavily on, is desirable. It is not completely clear how the data is input to the network.

Response: We thank the reviewer for this comment. We have expanded the description of the Ceograph and interaction-conditioned graph convolution, which is implemented using *conv.NNConv* method in pytorch.

(https://pytorch-geometric.readthedocs.io/en/latest/generated/torch_geometric.nn.conv.NNConv.html#torch_geometric.nn.conv.NNConv)

The description in the revised manuscript is copied below:

“Ceograph was constructed with three cell spatial interaction-conditioned graph convolutional (CSIGC) layers^[44] followed by a Subgroup Mean Pooling layer and a Softmax layer. The Ceograph consisted of nodes (nuclei) and edges (representing their spatial interactions). For each node, 11 nuclear morphological features were utilized as input data: nuclear area, convex area, eccentricity, extent, filled area, major axis length, minor axis length, the ratio of perimeter squared to area, perimeter, solidity, and prediction confidence. For each edge, three features were used: the categorical edge type determined by the cell types of both the starting and ending nodes (resulting in a total of $n \times n$ types for n cell types), nuclear parallelism, and nuclear closeness. The computation algorithm of a CSIGC layer is illustrated in **Figure 1C** and **Table 1**. The interaction-conditioned graph convolution makes the convolution operation conditioned on the spatial interactions of cell nuclei (i.e. edge attributes).”

[Ref 44]. Simonovsky M, Komodakis N, editors. Dynamic edge-conditioned filters in convolutional neural networks on graphs. Proceedings of the IEEE conference on computer vision and pattern recognition; 2017

- Authors employed a state-of-the-art method for nuclear identification and classification, HD-Staining. However, they did not mention any quality check of this method for the datasets of this paper. Did a pathologist review and validated predictions made by HD-Staining to make sure they were good enough?

Response: We thank the reviewer for this comment. Yes, our pathologists and co-authors of this manuscript (Drs. Justin Bishop, and Zhikai Chi) have reviewed and approved the HD-staining results for datasets from all three examples used in this study.

The HD-Staining method was originally developed and published in lung cancer tissues. Dr. Justin Bishop has reviewed the results in lung cancer tissues. In the HD-Staining study, the overall classification accuracy in lung cancer tissue was 85% in the testing set, while the accuracy for tumor nuclei was 90% in testing set. For our first example (the classification of lung ADC vs. SCC) and the third example (EGFR TK1 Ttx response prediction), lung cancer tissue images were used in both applications, the classification accuracy of cell nuclei should be around 85%.

Our recent publications showed the HD-Staining method can be adapted and applied in pathology image datasets of other solid cancer tissues, such as head and neck cancer, breast cancer, and lung cancer squamous cell carcinoma. For the second example, (the risk prediction of malignant transformation in Oral potentially malignant disorders), in the revision process, a pathologist, Dr. Zhikai Chi, reviewed and validated the cell classification results in oral tissues. Actually, classification of stratum corneum, stratum basale, other strata and non-epithelium in oral tissues is a much easier task comparing to classification of different cell types in cancer tissues.

Minor:

- The authors should tone down some sections of the paper where they claim that there is a need for computational methods capable of characterizing complex cell spatial organization. In fact, they claim to be the first paper to characterize cell spatial organization using graph models at single cell level. Although this could be true in the deep learning arena, there are multiple handcrafted approaches that have done so, see for example:

[10.1016/j.media.2020.101696](https://doi.org/10.1016/j.media.2020.101696), [10.1016/j.media.2020.101903](https://doi.org/10.1016/j.media.2020.101903) , [10.1038/s41698-022-00277-5](https://doi.org/10.1038/s41698-022-00277-5).

Response: We appreciate the reviewer's comment and acknowledge the significance of recognizing existing handcrafted approaches that have addressed cell spatial organization. In our revised manuscript, we have provided a more balanced overview of previous work, referencing and discussing earlier studies that characterized cell spatial organization. We have moderated our claim regarding the need for computational methods that characterize complex cell spatial organization and have removed the claim that ours is the first paper to represent cell spatial organization using graph models at the single-cell level.

In the Introduction Section:

“Although recent advances in modeling cell-cell spatial interactions and their impact on biomarker expressions are noteworthy¹¹, many studies have predominantly concentrated on the spatial distance between two cells. This focus, although informative, may neglect the more intricate facets of cell spatial architecture, such as distinct cell types, cell structure alignment, and interactions among multiple cells. Furthermore, a prevalent approach has been to amalgamate the influences of neighboring cells indiscriminately, potentially overlooking distinct cell-cell interactions. Recent handcrafted methods¹²⁻¹⁴ have delved into aspects of cell spatial organization. Moreover, SpaGCN¹⁵, a graph convolutional network, integrates gene expression, spatial location, and histology to discern spatial domains and spatially variable genes in spatial molecular profiling data. In this study, we developed a graph convolutional network to decipher intricate cellular interactions at the single-cell scale. It underscores the importance of computational methods that holistically understand cell spatial organization, emphasizing cell locations, proximities, and relationships among various cell types, such as tumor cells, stroma cells, and immune cells.”

[Ref 12]. Javed S, Mahmood A, Fraz MM, Koohbanani NA, Benes K, Tsang Y-W, Hewitt K, Epstein D, Snead D, Rajpoot N. Cellular community detection for tissue phenotyping in colorectal cancer histology images. *Medical Image Analysis*. 2020;63:101696.

[Ref 13]. Lu C, Koyuncu C, Corredor G, Prasanna P, Leo P, Wang X, Janowczyk A, Bera K, Lewis Jr J, Velcheti V, Madabhushi A. Feature-driven local cell graph (Flock): New computational pathology-based descriptors for prognosis of lung cancer and HPV status of oropharyngeal cancers. *Medical Image Analysis*. 2021;68:101903.

[Ref 14]. Ding R, Prasanna P, Corredor G, Barrera C, Zens P, Lu C, Velu P, Leo P, Beig N, Li H, Toro P, Berezowska S, Baxi V, Balli D, Belete M, Rimm DL, Velcheti V, Schalper K, Madabhushi A. Image analysis reveals molecularly distinct patterns of TILs in NSCLC associated with treatment outcome. *npj Precision Oncology*. 2022;6(1):33.

[Ref 15]. Hu J, Li X, Coleman K, Schroeder A, Ma N, Irwin DJ, Lee EB, Shinohara RT, Li M. SpaGCN: Integrating gene expression, spatial location and histology to identify spatial domains and spatially variable genes by graph convolutional network. *Nature Methods*. 2021;18(11):1342-51.

- Results of the research work should not be described in the last paragraph of the Introduction

Response: We left out the results of the research work from the last paragraph of the Introduction Section in the revised manuscript. We copy the last paragraph of the Introduction Section here for your convenience:

"We introduce Ceograph, a novel approach designed to decipher the complex interplay between cell morphologies and their spatial interactions, and its potential utility in diverse clinical contexts. With an increasing demand for methods that can characterize cell spatial organization and predict clinical outcomes using tissue images, Ceograph aims to bridge this gap. The versatility of this method is explored across three critical applications: 1) lung cancer subtype classification, 2) assessing the risk of malignant transformation in patients with oral potentially malignant disorders (OPMD), and 3) predicting treatment response in lung cancer. A unique strength of Ceograph lies in its potential for interpretability. For instance, in the context of lung cancer subtypes, we aim to see if Ceograph can capture the nuanced differences in cell organization between lung squamous cell carcinoma (SCC) and lung adenocarcinoma (ADC). Similarly, with OPMD patients, it might be possible to discern if disruptions in epithelial strata structure have implications for malignant transformations. Lastly, for lung cancer treatment response, understanding cellular features such as tumor nuclei morphology and stroma-stroma interactions could offer insights into treatment sensitivities. This work represents the first study to develop interpretable deep cell spatial interaction learning models for clinical outcome prediction and identification of cell spatial organization features that predict clinical outcomes using tissue images."

REVIEWER COMMENTS

Reviewer #1 (Remarks to the Author):

I would like to commend the authors on their thoughtful and detailed response to my comments. The revised manuscript, figures and supplementary materials form an impactful publication that is a meaningful contribution to the field of computational tissue image analysis. I recommend this manuscript for publication in Nature Communications.

Reviewer #2 (Remarks to the Author):

I want to thank the authors for taking the time to address my comments. I consider the paper to be clearer now. I just have a few minor comments:

1. I appreciate the clarifications and adjustments authors made to the paper regarding the prognostic analysis. However, there is something that was not explained. Why Dr. Anderson selected 68.5 as threshold? Is this a standard cut off for patients with lung cancer?
2. With regards to the predictive analysis, I thank the authors for explaining explained that 31 was set as the threshold because it was the *median overall survival*. This should be explicitly mentioned in the paper as the statement "The OS cutoff was selected to ensure non-benefitting and benefitting groups are relatively balanced" is slightly vague in this sense.
3. Authors are encouraged to include the details of the validation of nuclear segmentation and classification performed by pathologists, as mentioned in the letter, in the manuscript or supplementary materials.

The reviewers' comments verbatim are in blue. Our responses are in plain black text. Our corresponding changes were highlighted in red in the manuscript.

REVIEWER COMMENTS

Reviewer #1 (Remarks to the Author):

I would like to commend the authors on their thoughtful and detailed response to my comments. The revised manuscript, figures and supplementary materials form an impactful publication that is a meaningful contribution to the field of computational tissue image analysis. I recommend this manuscript for publication in Nature Communications.

Response: We deeply appreciate the time and effort you dedicated to reviewing our work. We are honored to hear that you recommend our manuscript for publication in Nature Communications. Once again, thank you for your insightful feedback and support throughout the revision process.

Reviewer #2 (Remarks to the Author):

I want to thank the authors for taking the time to address my comments. I consider the paper to be clearer now. I just have a few minor comments:

Response: Thank you for your kind words and positive feedback on our revised manuscript.

1. I appreciate the clarifications and adjustments authors made to the paper regarding the prognostic analysis. However, there is something that was not explained. Why Dr. Anderson selected 68.5 as threshold? Is this a standard cut off for patients with lung cancer?

Response: Thank you for highlighting this point and allowing us to clarify our methodology. In the transformation of OPMD to malignancy, patients with OPMD who do not progress to oral cancer within a 5-year period are typically considered low-risk. Therefore, when Dr. Curtis Pickering was selecting samples of low-risk patients, his goal was to identify those who did not develop oral cancer for at least 60 months which is a clinically important cutoff. Among the 17 patients who were selected as low risk patients by Dr. Pickering, the one closest to this clinical important cutoff criterion had a follow-up time of 68.5 months. Therefore, we used this data-driven cutoff, 68.5 months, in the paper. We concur that our original explanation was not clear. As such, we have revised the description in the Methods Section to reflect the intended cutoff value of 60 months based on clinical significance. There are no different in analysis and results when using 60 or 68.5 months.

In the Methods Section (sub-section “Dataset”):

“In this study, for OPMD, patients who do not develop oral cancer within 60 months are categorized as low-risk, while those who develop oral cancer within 42 months are considered high-risk. Based on this criterion, and in conjunction with the set exclusion criteria, we identified 17 patients who had a follow-up duration exceeding 60 months without developing oral cancer (the shortest follow-up time among these patients was 68.5 months). On the other hand, 6 patients who developed oral cancer within a span of 42 months (with CFS < 42 months and an event status of true) were classified as high-risk and included in the OPMD1 training data.”

2. With regards to the predictive analysis, I thank the authors for explaining explained that 31 was set as the threshold because it was the *median overall survival*. This should be explicitly mentioned in the paper as the statement “The OS cutoff was selected to ensure non-benefitting and benefitting groups are relatively balanced” is slightly vague in this sense.

Response: We appreciate your feedback and agree that our initial description was somewhat vague. We have revised the description in the Methods Section as you suggested. The revised content is provided below for your convenience.

In the Methods Section (the second paragraph of sub-section “Ceograph for EGFR TKI Ttx Response Prediction”):

“The cutoff was selected based on the median OS time, which is 31.2 months, to ensure that the non-benefitting and benefitting groups are relatively balanced.”

3. Authors are encouraged to include the details of the validation of nuclear segmentation and classification performed by pathologists, as mentioned in the letter, in the manuscript or supplementary materials.

Response: We changed the description in the Methods Section as you suggested. The revised content is provided below for your convenience.

In the Methods Section (the last paragraph of sub-section “Nuclei Segmentation and Classification Using HD-Staining”):

“Board-certified clinical pathologists have reviewed and approved the HD-staining results for datasets from all three examples used in this study.”

REVIEWERS' COMMENTS

Reviewer #2 (Remarks to the Author):

I want to thank the authors for addressing my comments. I don't have additional feedback.

The reviewers' comments verbatim are in blue. Our responses are in plain black text.

REVIEWER COMMENTS

Reviewer #2 (Remarks to the Author):

I want to thank the authors for addressing my comments. I don't have additional feedback.

Response: We deeply appreciate the time and effort you dedicated to reviewing our work. We are honored to hear that you recommend our manuscript for publication in Nature Communications. Once again, thank you for your insightful feedback and support throughout the revision process.